# CFO: Learning Continuous-Time PDE Dynamics via Flow-Matched Neural Operators

**Xianglong Hou**
Graduate Group in Applied Mathematics and Computational Science
University of Pennsylvania
Philadelphia, PA 19104
`xlhou@sas.upenn.edu`

**Xinquan Huang, Paris Perdikaris**
Department of Mechanical Engineering and Applied Mechanics
University of Pennsylvania
Philadelphia, PA 19104
`{huang26,pgp}@seas.upenn.edu`

## Abstract

Neural operator surrogates for time-dependent partial differential equations (PDEs) conventionally employ autoregressive (AR) prediction schemes, which accumulate error over long rollouts and require uniform temporal discretization. We introduce the **Continuous Flow Operator (CFO)**, a framework that learns continuous-time PDE dynamics without the computational burden of standard continuous approaches, e.g., neural ODE. The key insight is repurposing flow matching to directly learn the right-hand side of PDEs without backpropagating through ODE solvers. CFO fits temporal splines to trajectory data, using finite-difference estimates of time derivatives at knots to construct probability paths whose velocities closely approximate the true PDE dynamics. A neural operator is then trained via flow matching to predict these analytic velocity fields. This approach is inherently time-resolution invariant: training accepts trajectories sampled on trajectory-specific, non-uniform time grids while inference queries solutions at any temporal resolution through ODE integration. Across four benchmarks (Lorenz, 1D Burgers, 2D diffusion-reaction, 2D shallow water), CFO demonstrates superior long-horizon stability and remarkable data efficiency. CFO trained on only 25% of irregularly subsampled time points outperforms autoregressive baselines trained on complete data, with relative error reductions up to 87%. Despite requiring numerical integration at inference, CFO achieves competitive efficiency, exceeding AR accuracy using only half the AR rollout step budget, while uniquely enabling reverse-time inference and arbitrary temporal querying. Code is available at  github.com/shannon-hou/CFO_official.

## 1 Introduction

Time-dependent partial differential equations (PDEs) are fundamental to modeling dynamical phenomena across the physical, biological, and engineering sciences. Neural PDE solvers have emerged as a promising paradigm for learning mappings between function spaces and fast simulation of complex PDEs (Azizzadenesheli et al., 2024). However, current approaches face critical limitations: autoregressive (AR) methods suffer from error accumulation over long horizons and require uniform time grids (Sanchez-Gonzalez et al., 2020); spatio-temporal methods scale poorly with the space-time volume and struggle with causality (Wang et al., 2021); continuous approaches like Neural ODEs (Chen et al., 2018) require expensive backpropagation through ODE solvers.

In this study, we propose the **Continuous Flow Operator** (CFO), a continuous-time neural framework that circumvents these limitations through a key insight: repurposing flow match-

ing (Lipman et al., 2022; Albergo et al., 2023) typically used for generative modeling, to learn continuous-time dynamics without backpropagating through ODE solvers.

We directly learn the right-hand side dynamics of a PDE system by matching it with the analytic velocity of spline-based interpolants over trajectories. In doing so, CFO can be trained on arbitrary, non-uniform time grids to fully exploit available temporal data and queried at any temporal resolution during inference, while retaining the training efficiency of discrete methods.

Our contributions are summarized as:

- **A continuous-time framework for learning PDE dynamics.** We propose a novel method that learns the inherent continuous-time right-hand side (RHS) by training a neural operator to match the analytic velocity (i.e., the time derivative) of spline-based interpolants, thereby avoiding costly backpropagation through ODE solvers via direct derivative supervision.
- **Time-resolution invariance.** The proposed framework inherently handles irregular, trajectory-specific time grids during training and supports arbitrary-resolution inference, which is unattainable with standard autoregressive methods.
- **State-of-the-art performance with minimal data.** Across four benchmarks, CFO achieves superior long-horizon stability; models trained on 25% irregular subsamples outperform full-data autoregressive baselines, with relative error reductions up to 87%.

## 2 RELATED WORK

**Neural PDE Solvers and Operator Learning.** Neural operator learning has emerged as a powerful paradigm for solving PDE families by learning mappings between function spaces. Pioneering frameworks include DeepONet (Lu et al., 2021) with branch-trunk architectures and FNO (Li et al., 2020a) with spectral convolutions. Recent extensions handle irregular geometries (Geo-FNO (Li et al., 2023), multipole graph operators (Li et al., 2020b)), multi-scale structures (WNO (Tripura & Chakraborty, 2022)), improved depth (U-NO (Rahman et al., 2022)), and irregular sampling (Galerkin Transformer (Cao, 2021), CViT (Wang et al., 2024)). For time-dependent PDEs, autoregressive prediction suffers from exposure bias and cumulative error, while treating time as a spatio-temporal coordinate scales poorly and complicates causality enforcement (Wang et al., 2021).

**Neural ODEs for Continuous-time Modeling.** Neural ODEs (Chen et al., 2018) and variants (Kidger et al., 2020; Dupont et al., 2019) provide continuous-time modeling for PDE systems (Sholokhov et al., 2023; Wen et al., 2023). While theoretically elegant, these methods require backpropagation through ODE solvers via the adjoint method, remaining computationally expensive and memory-intensive even when operating in latent spaces.

**Generative Models for PDE Simulation.** Recent generative approaches treat solution fields as samples from learned distributions. DiffusionPDE (Huang et al., 2024) uses EDM-style diffusion (Karras et al., 2022) with physics guidance during sampling; CoCoGen (Jacobsen et al., 2025) extends score-based models (Song et al., 2020) with physically-consistent sampling; PBFM (Baldan et al., 2025) integrates physics directly into flow matching (Lipman et al., 2022) training via conflict-free gradient updates. While effective, these methods typically require explicit PDE expressions and suffer from computational overhead because of multi-step sampling and space-time generation.

**Our Approach.** CFO bridges continuous-time modeling with computational efficiency through a key innovation: using flow-matching objectives to learn continuous-time vector fields directly from trajectory derivatives, avoiding the costly ODE solver backpropagation required by Neural ODEs. Unlike generative PDE solvers that need explicit differential operators, CFO implicitly encodes dynamics through probability paths aligned with PDE evolution, maintaining temporal causality without requiring governing equations.

Three advantages distinguish our framework: (1) *Model agnosticism* – compatible with any neural operator architecture (e.g., FNO, U-Net (Ronneberger et al., 2015)); (2) *Time-resolution invariance* – handles irregular, trajectory-specific time grids during training while supporting arbitrary-resolution inference; (3) *Computational efficiency* – achieves autoregressive training speeds while maintaining continuous-time benefits.

While Zhang et al. (2024) apply flow matching with piecewise linear interpolation to clinical time series and treat trajectories as generic sequences, CFO instead exploits PDE structure: by fitting high-order splines with finite-difference derivative estimates at knots, we construct probability paths whose velocities closely match true PDE dynamics. This physics-aware design keeps the learned flow in high-probability regions consistent with governing equations, enabling accurate predictions even when trained on only 25% of irregularly sampled time points.

## 3 PRELIMINARIES

**Method of Lines.** The method of lines (MOL) is a numerical method for solving time-dependent PDEs. The core idea is first to discretize the spatial domain, transforming the PDE into a large system of ODEs in time, which can then be solved using a standard numerical integrator. Formally, after spatial discretization (e.g., on a grid), the PDE in Eq. (2) becomes a system of ODEs:

$$\frac{d}{dt}u_h(t) = \mathcal{N}_h(u_h(t)), \quad u_h(0) = u_{0,h},$$

where $u_h(t)$ is a vector representing the solution on the spatial grid at time $t$, and $\mathcal{N}_h$ is a finite-dimensional approximation of the spatial operator $\mathcal{N}$ (e.g., via finite differences or spectral methods). However, when $\mathcal{N}$ is unknown or difficult to specify, instead of relying on a fixed, predefined discretization $\mathcal{N}_h$, we learn a neural operator $\mathcal{N}_\theta$ that directly approximates the continuous-time dynamics from data. This allows us to define a continuous-time ODE system that can be integrated with arbitrary time steps, inheriting the flexibility of MOL while avoiding the limitations of discrete-time autoregressive models.

While MOL provides a conceptual basis for learning a continuous-time operator $\mathcal{N}_\theta$, a naive implementation that matches solution trajectories would require backpropagation through an ODE solver, which is computationally expensive, as seen in Neural ODEs (Chen et al., 2018).

Generative modeling frameworks like flow matching (Lipman et al., 2022) and stochastic interpolants (Albergo et al., 2023) have put forth an efficient alternative. These methods learn a continuous-time vector field by directly regressing it against the velocity of a predefined probability path connecting two distributions. This approach avoids costly ODE integration during training, a key feature we leverage in CFO. Flow matching is a special case of stochastic interpolants with no noise term. We briefly review these concepts below.

**Stochastic Interpolants and Flow Matching.** Stochastic interpolants (Albergo et al., 2023) and flow matching (Lipman et al., 2022) provide a framework for learning a continuous-time vector field that transports one probability distribution to another. The core idea is to define a probability path $p_t$ that connects a source distribution $p_0$ to a target $p_1$. A stochastic interpolant is typically defined as $I_t = s(t, X_0, X_1) + \gamma(t)Z$, where $X_0, X_1$ are drawn from a probability measure $q(x_0, x_1)$, $Z \sim \mathcal{N}(0, I)$ is independent noise, and the terms satisfying boundary conditions:

$$s(0, x_0, x_1) = x_0, \quad s(1, x_0, x_1) = x_1, \quad \gamma(0) = \gamma(1) = 0.$$

The key insight is that the vector field $v(t, x) = \mathbb{E}[\frac{d}{dt}I_t | I_t = x]$ that generates this path can be learned directly by regressing a neural network $v_\theta(t, x)$ against the analytic velocity $\frac{d}{dt}I_t$ of the predefined process:

$$\mathcal{L}(\theta) = \mathbb{E}_{t,x_0,x_1,z}\left[\|v_\theta(t, I_t) - \frac{d}{dt}I_t\|^2\right]. \tag{1}$$

This regression objective avoids costly ODE integration during training. During inference, the learned vector field $v_\theta$ defines an ODE, $\frac{d}{dt}X_t = v_\theta(t, X_t)$, which can be solved with a numerical integrator to map samples from $p_0$ to $p_1$. CFO adapts this principle by constructing a more sophisticated probability path based on splines that incorporate derivative information and interpolate entire solution trajectories, rather than only matching the endpoints.

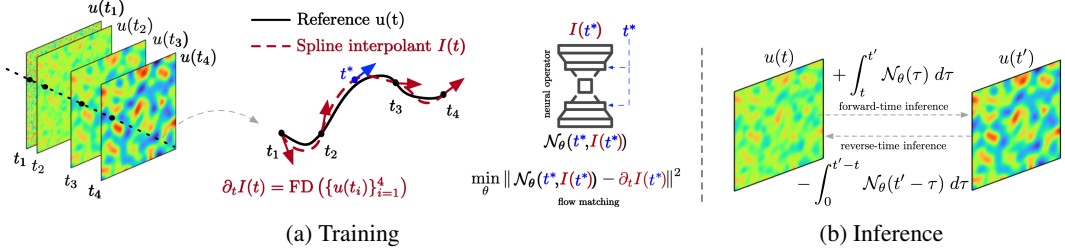

(a) Training                    (b) Inference

Figure 1: Overview of the Continuous Flow Operator (CFO) framework. (a) **Training: flow matching on a spline path.** For each trajectory with snapshots $\{u(t_i)\}$, we fit a temporal spline $s(t)$ that interpolates the data and matches finite-difference derivative estimates at knots. The spline's analytic time derivative $\partial_t s(t)$ provides exact velocity targets for training a neural operator $\mathcal{N}_\theta$ via the flow matching objective—no ODE integration required during training. (b) **Inference: continuous-time rollout.** The trained operator $\mathcal{N}_\theta(t, u)$ defines a continuous vector field $\dot{u}_\theta = \mathcal{N}_\theta(t, u_\theta)$. Given initial condition $u(t)$, we compute $u(t')$ by numerical integration; reverse-time inference integrates backward.

## 4  CONTINUOUS FLOW OPERATOR

### 4.1  METHODOLOGY

Consider a time-dependent PDE on a domain $[0, T] \times \Omega \subset \mathbb{R} \times \mathbb{R}^d$:

$$\partial_t u(t, x) = \mathcal{N}\big(u(t, x)\big), \qquad u(0, x) = u_0(x), \quad x \in \Omega,\ t \in [0, T], \tag{2}$$

where $\mathcal{N}$ is an unknown spatial differential operator and $u_0(x)$ is the initial condition.

**Notation.** We suppress spatial coordinates and write $u(t) \equiv u(t, \cdot) \in \mathcal{V}$. For sampled times $\mathcal{T} := (t_i)_{i=0}^{N}$ we write the restriction $u|_\mathcal{T} := \big(u(t_0), \dots, u(t_N)\big)$. When $\mathcal{T}$ is clear from context, we abbreviate the snapshot vector to $\mathbf{u} := u_\mathcal{T}$.

**Problem Statement.** Let $M$ trajectories $\{u^j(t)\}_{j=1}^{M}$ be generated from initial conditions $\{u_0^j\}_{j=1}^{M}$. For each trajectory $j$, we observe snapshots at (possibly irregular) times $\mathcal{T}^j := (t_i^j)_{i=0}^{N_j}$ and write $\mathbf{u}^j := u^j|_{\mathcal{T}^j} = \big(u^j(t_0^j), \dots, u^j(t_{N_j}^j)\big)$. Given the dataset $\{\mathbf{u}^j\}_{j=1}^{M}$, our goal is to learn an operator that maps from any initial condition $u_0$ to the solution trajectory $u(t)$ for any $t \in [0, T]$. For exposition, we drop the superscript $j$ and set the horizon to $T = 1$.

**Probability Path Formulation.** We define a temporal spline $s(t; \mathbf{u})$ that interpolates the snapshots:

$$s(t_i; \mathbf{u}) = u(t_i), \qquad i = 0, \dots, N. \tag{3}$$

The (multi-point) stochastic interpolant is

$$I(t; \mathbf{u}) = s(t; \mathbf{u}) + \gamma(t)\, z, \qquad t \in [0, 1], \tag{4}$$

with boundary conditions $\gamma(t_i) = 0$ for $i = 0, \dots, N$ and $z \sim \mathcal{N}(0, I)$, independent of $\mathbf{u}$. In the MOL one discretizes space to obtain $\mathcal{N}_h \approx \mathcal{N}$ with $\partial_t u(t) = \mathcal{N}_h\big(u(t)\big)$. When $\mathcal{N}$ is unavailable, we take a data-driven route: (i) estimate $\partial_t u(t_i)$ from $\mathbf{u}$ using finite-difference stencils along time; (ii) choose the spline $s(t; \mathbf{u})$ so that its derivatives at the knots match these estimates. This allows us to control the derivative accuracy via the stencil order and the path smoothness via the spline degree.

**Finite Difference Approximation.** The order of finite-difference stencils depends on the number of points used (Strikwerda, 2004). Below are examples for the first and second time derivatives. Let $d_i \approx \partial_t u(t_i)$ and $a_i \approx \partial_{tt} u(t_i)$ denote derivative estimates at time $t_i$. Define the steps $h_i := t_{i+1} - t_i$ and $\Delta t := \max_i h_i$.

A *two-point stencil* is defined as $d_i = \frac{u(t_{i+1}) - u(t_i)}{h_i}$, which is first-order accurate: $\partial_t u(t_i) = d_i + O(\Delta t)$. A *three-point stencil* provides second-order accuracy for the first derivative and first-order

accuracy for the second derivative on irregular grids. For an interior point $t_i$,

$$d(t_i) = -\frac{h_i}{h_{i-1}(h_{i-1} + h_i)} u(t_{i-1}) + \frac{h_i - h_{i-1}}{h_{i-1}h_i} u(t_i) + \frac{h_{i-1}}{h_i(h_{i-1} + h_i)} u(t_{i+1}),$$
$$a(t_i) = \frac{2}{h_{i-1} + h_i} \left( \frac{u(t_{i+1}) - u(t_i)}{h_i} - \frac{u(t_i) - u(t_{i-1})}{h_{i-1}} \right). \tag{5}$$

These yield $\partial_t u(t_i) = d_i + O(\Delta t^2)$ and $\partial_{tt} u(t_i) = a_i + O(\Delta t)$. Endpoint estimates can be obtained with standard one-sided stencils. Higher-order accuracy is achievable with more points (see Appendix A.1.1 for details).

**Segment Interpolation.** With the derivative estimates at the knots, we construct the spline $s(t; \mathbf{u})$ segment-wise. On each interval $[t_i, t_{i+1}]$, we use a polynomial that matches the values and estimated derivatives at the endpoints. This approach, known as Hermite interpolation, allows us to control the spline's smoothness and the accuracy of its derivatives at the knots. For a spline to be $C^s$ continuous, we must impose conditions on derivatives up to order $s$ at each endpoint, which requires a polynomial of degree at least $2s + 1$ on each segment.

**Representative Cases.** We consider two cases: linear splines ($C^0$) and quintic splines ($C^2$).

LINEAR SPLINE. The simplest choice is a *linear* spline, which only interpolates the values $u(t_i)$ and $u(t_{i+1})$. On each segment $t \in [t_i, t_{i+1}]$, the spline is:

$$s(t; \mathbf{u}) = \frac{t_{i+1} - t}{h_i} u(t_i) + \frac{t - t_i}{h_i} u(t_{i+1}). \tag{6}$$

The resulting spline is continuous but not differentiable at the knots. Its one-sided derivative at $t_i$ matches the first-order forward-difference estimate, yielding $\partial_t s(t_i; \mathbf{u}) = \mathcal{N}(u(t_i)) + O(\Delta t)$.

QUINTIC SPLINE. To incorporate higher-order derivative information, we use a *quintic* spline. On each segment, this spline matches the values $(u_{t_i}, u_{t_{i+1}})$, first derivatives $(d_{t_i}, d_{t_{i+1}})$, and second derivatives $(a_{t_i}, a_{t_{i+1}})$ at both endpoints. With the three-point finite difference stencils from Eq. (5), the spline derivatives at the knots satisfy:

$$\partial_t s(t_i; \mathbf{u}) = \mathcal{N}(u(t_i)) + O(\Delta t^2), \quad \partial_{tt} s(t_i; \mathbf{u}) = \partial_t \mathcal{N}(u(t_i)) + O(\Delta t).$$

This construction yields a globally $C^2$ smooth path that more accurately reflects the underlying dynamics. The explicit form is provided in Appendix A.1.2.

Unlike global methods such as natural splines, our segment-wise Hermite interpolation yields closed-form coefficients. It also allows for flexible choices of basis functions, smoothness, and derivative accuracy based on the physics prior. Compared to low-order splines, the higher-order spline here (also with high-order assigned derivatives) benefits from more accurate derivative approximations and higher smoothness: (i) high-order stencils make the probability flow match the exact flow of the physical dynamics as closely as possible, (ii) greater smoothness improves training stability and long-horizon accuracy. We show these effects in Sec. 5.1.

In practice, we use quintic splines by default, since they naturally capture acceleration, which appears in many physical laws (e.g., Newton's second law, wave equations). We do not go beyond quintic because of diminishing returns: in CFO, the total error comes from both the spline approximation and the neural operator training, and beyond quintic we observe only minor improvements in derivative accuracy while the network error dominates (see Sec. A.1.3 for detailed discussion).

For the noise term, we can also impose regularity on $\gamma(t)$ with splines. For example, $\gamma(t) = \gamma_0 \frac{(t-t_k)^m (t_{k+1}-t)^m}{(t_{k+1}-t_k)^{2m}}$, $t_k \le t < t_{k+1}$ with some constant $\gamma_0 > 0$ ensures $C^{m-1}$ continuity and vanishing noise at the knots. A detailed study can be found in Sec. 5.2.

## 4.2 TRAINING AND INFERENCE

**Training.** We learn a time-dependent neural operator $\mathcal{N}_\theta(t, u(t))$ that matches the velocity of the spline-based stochastic interpolant $I(t; \mathbf{u})$. Similar to the training objective in stochastic interpolants, the training loss is

$$\mathcal{L}(\theta) = \mathbb{E}_{\mathbf{u}, t, Z} \left[ \left\| \mathcal{N}_\theta(t, I(t; \mathbf{u})) - \partial_t I(t; \mathbf{u}) \right\|^2 \right], \tag{7}$$

where $t \sim \mathrm{Unif}[0,1]$, $\mathbf{u}$ denotes a sampled trajectory, $Z$ is the Gaussian noise independent of $\mathbf{u}$. $I(t; \mathbf{u})$ is the corresponding stochastic interpolant, and $\partial_t I(t; \mathbf{u})$ is its analytic spline derivative. This avoids backpropagating through an ODE solver.

**Inference.** After training, $\mathcal{N}_\theta$ defines a continuous-time right-hand side. For a new initial condition $u_0$, we predict by integrating

$$\frac{d}{dt} u_\theta(t) = \mathcal{N}_\theta(t, u_\theta(t)), \qquad u_\theta(0) = u_0, \tag{8}$$

with a standard ODE solver; by default, we use 4th-order Runge-Kutta (RK4). See Sec. 5.2 for details of solver choice and the number of function evaluations (NFE).

**Time-Resolution Invariance.** CFO is agnostic to time grids: (i) *Training* accepts trajectories with trajectory-specific and irregular time stamps. (ii) *Inference* queries (8) at arbitrary time resolutions or schedules (including times unseen in training). This allows mixing dense and sparse sequences during training and evaluating on any target grid at test time.

**Reverse-time Inference.** Unlike autoregressive next-step predictors, our learned time-dependent vector field induces a locally invertible flow map under standard Lipschitz conditions. This makes CFO suitable for inverse tasks such as inferring earlier states from a state observed at time $t_\star$. Given a complete state $u(t_\star)$ at any time $t_\star \in [0,1]$, one can recover the state at earlier times $s < t_\star$ by integrating the learned flow $\mathcal{N}_\theta(t, u(t))$ backward:

$$u_\theta(s) = u(t_\star) + \int_{t_\star}^{s} \mathcal{N}_\theta(\tau, u_\theta(\tau))\, d\tau = u(t_\star) - \int_{0}^{t_\star - s} \mathcal{N}_\theta\big(t_\star - \tau, u_\theta(t_\star - \tau)\big)\, d\tau,$$

We empirically validate this reverse-time inference capability on the dissipative Burgers' equation in Sec. 5.2.

**Probability-flow Transport.** For a fixed (possibly irregular) time grid, if $\mathcal{N}_\theta$ minimizes the loss function (7), the ODE (8) transports the distribution across the sequence of data marginals at each grid time. It generalizes the two-marginal result of Lipman et al. (2022); Albergo et al. (2023). (see Proposition A.1-A.2) for details. Empirical results also show that the induced probability flow closely tracks the true dynamics even with heterogeneous time grids across trajectories, especially for quintic CFO.

## 5 EXPERIMENTS

In this section, we evaluate the proposed CFO framework across four canonical benchmarks: Lorenz, 1D Burgers', 2D diffusion–reaction, and 2D shallow water equations. Our primary focus is on long-horizon rollouts under irregular subsampling, where we compare CFO against autoregressive baselines trained with teacher forcing. In addition, we perform ablation studies to assess the impact of the noise schedule, inference efficiency, long-horizon temporal extrapolation, and backbone architectures. All time series are normalized to the range $[0, 1]$. More results are shared in the Appendix A.4.

### 5.1 MAIN RESULTS

We evaluate CFO's time-resolution invariance by training on irregularly subsampled time grids and testing on the full-resolution grid. Models are trained using per-trajectory random time grids with keep ratios of 100%, 50%, and 25%. Autoregressive baselines, which require uniform time steps, are trained only on the full (100%) grid; for autoregressive, we try multiple architectures (including the same backbone as CFO) and report the best. All models learn the mapping from the initial condition to the entire trajectory. Table 1 reports the rollout accuracy (relative $L_2$ error, mean ± sd). Notably, Quintic CFO trained on just 25% of irregularly sampled time points still outperforms AR models trained on full-resolution data, achieving relative error reductions of 24.6%, 78.7%, 87.4%, and 82.8% on the four benchmarks, respectively. This highlights the CFO's ability to leverage all available data, regardless of sampling density or regularity, and to generalize effectively to arbitrary time resolutions.

Table 1: Time-resolution invariant results across four benchmarks: **CFO is robust to irregular time subsampling.** Training uses per-trajectory random time grids at keep ratios 100%, 50%, and 25%; Testing is at full resolution. Values are relative $L_2$ error↓ (mean ± sd). Bold indicates the best result in each sub-row; "–" means AR not suitable for irregular grids.

| Dataset | Sampling | Autoregressive | Linear CFO | Quintic CFO |
|---------|----------|----------------|------------|-------------|
| Lorenz | 100% | $(9.04 \pm 2.80) \times 10^{-2}$ | $(6.42 \pm 0.69) \times 10^{-2}$ | $\mathbf{(4.53 \pm 0.68) \times 10^{-2}}$ |
| | 50% | – | $(7.85 \pm 0.25) \times 10^{-2}$ | $\mathbf{(4.75 \pm 0.78) \times 10^{-2}}$ |
| | 25% | – | $(9.39 \pm 0.77) \times 10^{-2}$ | $\mathbf{(6.82 \pm 0.48) \times 10^{-2}}$ |
| Burgers | 100% | $(3.34 \pm 0.14) \times 10^{-2}$ | $\mathbf{(5.75 \pm 0.72) \times 10^{-3}}$ | $(5.89 \pm 0.79) \times 10^{-3}$ |
| | 50% | – | $(8.91 \pm 1.30) \times 10^{-3}$ | $\mathbf{(6.97 \pm 0.73) \times 10^{-3}}$ |
| | 25% | – | $(1.04 \pm 0.15) \times 10^{-2}$ | $\mathbf{(7.09 \pm 1.10) \times 10^{-3}}$ |
| DR | 100% | $(4.23 \pm 0.59) \times 10^{-1}$ | $\mathbf{(4.35 \pm 0.67) \times 10^{-2}}$ | $(4.37 \pm 0.47) \times 10^{-2}$ |
| | 50% | – | $(6.88 \pm 0.49) \times 10^{-2}$ | $\mathbf{(6.19 \pm 0.35) \times 10^{-2}}$ |
| | 25% | – | $(7.25 \pm 0.48) \times 10^{-2}$ | $\mathbf{(5.32 \pm 0.65) \times 10^{-2}}$ |
| SWE | 100% | $(9.04 \pm 0.36) \times 10^{-2}$ | $(5.93 \pm 0.95) \times 10^{-3}$ | $\mathbf{(4.56 \pm 0.38) \times 10^{-3}}$ |
| | 50% | – | $(7.64 \pm 0.52) \times 10^{-3}$ | $\mathbf{(6.57 \pm 0.66) \times 10^{-3}}$ |
| | 25% | – | $(1.69 \pm 0.38) \times 10^{-2}$ | $\mathbf{(1.55 \pm 0.22) \times 10^{-2}}$ |

**Lorenz System.** We first consider the Lorenz system (Lorenz, 2017), a 3D chaotic ODE with spatial-derivative-free dynamics. In this simple setting, the state is a vector $u(t) \in \mathbb{R}^3$ representing a system of ODEs rather than a PDE with spatial dependence:

$$\dot{x} = \sigma(y - x), \quad \dot{y} = x(\rho - z) - y, \quad \dot{z} = xy - \beta z,$$

with $(\sigma, \rho, \beta) = (10, 28, 8/3)$. We use an MLP for both CFOs and baseline autoregressive methods.

Since the Lorenz vector field is derivative-free, we directly evaluate how well the spline-implied velocity $s(t; \mathbf{u})$ approximates the true dynamics. We analyze this by regularly downsampling trajectories to resolutions of $2\Delta t$ and $4\Delta t$. Figure 2 plots the average relative $L_2$ error across a normalized trajectory segment. For a fixed step size, the quintic spline achieves uniformly lower error. The linear spline is only comparable near the segment's midpoint, where its secant slope approximates a central difference; its error increases near endpoints, where the stencil becomes one-sided. As $\Delta t$ decreases, both splines improve, but the quintic spline converges faster. Table 4 confirms these convergence rates at the endpoints: the linear spline is first-order accurate ($\mathcal{O}(\Delta t)$), while the quintic spline is second-order ($\mathcal{O}(\Delta t^2)$), matching finite-difference theory.

In addition, we study the training stability and long-horizon accuracy under irregular subsampling. The spline-implied velocity error (Figure 2) reflects how closely the induced flow aligns with the true dynamics; lower error ensures the learned flow remains in higher-probability regions. Thus, the high-order spline has two advantages: (i) it improves long-horizon accuracy and is more data-efficient. When only a fraction of snapshots is available, the Quintic CFO maintains low rollout error and degrades more gracefully than the Linear CFO, demonstrating stronger robustness to data scarcity under irregular subsampling (Table 1). (ii) It enhances training stability. As shown in Figure 3, Quintic CFO achieves lower training loss and faster convergence compared to Linear CFO, particularly in tasks with irregular time sampling.

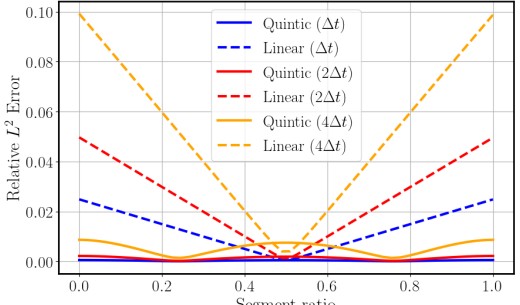

Figure 2: Spline fitting error on the Lorenz system. Average velocity-field approximation error for quintic and linear splines at different time-step sizes ($\Delta t, 2\Delta t, 4\Delta t$), with $\Delta t = 0.005$s.

**1D Burgers' Equation.** We test the CFO on the 1D Burgers' equation, a standard benchmark for nonlinear dynamics. The setup follows Wang et al. (2021). CFO uses a 1D U-Net, while the

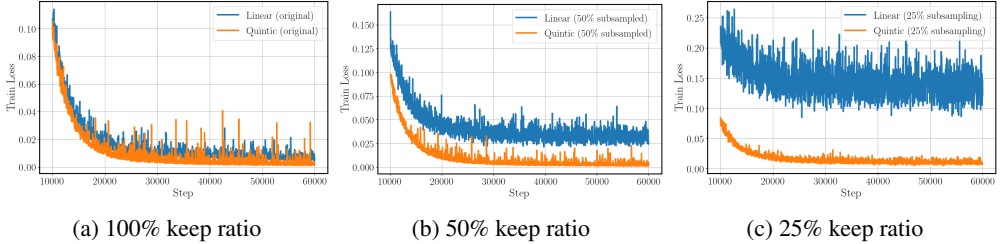

(a) 100% keep ratio · (b) 50% keep ratio · (c) 25% keep ratio

Figure 3: Training comparison between linear and quintic CFO on the Lorenz system. Training loss (10k–60k iterations) for linear and quintic CFO under different keep ratios: (a) 100%, (b) 50%, (c) 25%.

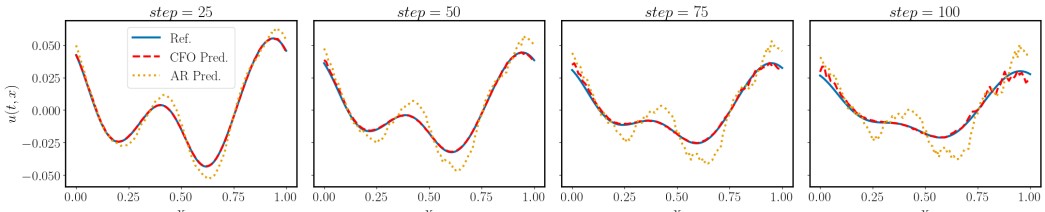

Figure 4: Worst-case prediction for Burgers' equation. Quintic CFO (trained on 25% irregular data) accurately predicts the full trajectory for the test sample with the highest error. Shown from left to right: solutions at the 25, 50, 75, and 100th time steps.

autoregressive baseline is an MLP, which performed best among several architectures tested (U-Net, MLP, FNO, see Appendix A.4.5).

As shown in Table 1, Quintic CFO trained on just 25% of irregularly sampled time points achieves a 78.7% relative error reduction compared to the autoregressive baseline trained on full data. Figure 4 visualizes the rollout for the test sample with the highest error (0.0307 relative $L_2$ error); even in this worst-case scenario, the prediction remains highly accurate.

**2D Diffusion-Reaction Equation.** We evaluate CFO on the 2D diffusion-reaction (DR) benchmark from PDEBench (Takamoto et al., 2022). We use a 2D U-Net for both CFO and the autoregressive baseline. As shown in Table 1, CFO significantly outperforms the autoregressive model. Quintic CFO trained on just 25% of irregularly sampled time points achieves an 87.4% relative error reduction compared to the autoregressive baseline trained on full data. Figure 5 visualizes a long-horizon rollout, showing that CFO accurately captures the complex pattern evolution, while the autoregressive model quickly diverges.

**2D Shallow Water Equation.** We evaluate CFO on the 2D shallow water equations (SWE) from PDEBench (Takamoto et al., 2022). CFO uses a 2D U-Net; the strongest autoregressive baseline uses a Diffusion Transformer (DiT) architecture. As shown in Table 1, quintic CFO (with 25% data keep ratio) attains 82.8% lower relative error than the autoregressive model trained on the full dataset. Figure 6 compares predictions over time and shows that, even on its worst sample, CFO substantially outperforms the autoregressive model.

## 5.2 ABLATION STUDY

**Impact of ODE solvers and function evaluations.** We study how CFO's inference accuracy and computational cost vary with the ODE solver and the number of function evaluations (NFE). We compare Euler, Heun, and RK4. Experiments show that higher-order numerical integrators achieve lower error for a fixed NFE. Table 2 fixes the solver to RK4; additional solver sweeps are provided in Tables 8 and 9. CFO's continuous-time formulation supports flexible solver choice and NFE budgets, unlike autoregressive (AR) rollouts that use fixed steps. For RK4, across all benchmarks we find that: with only 50% of the AR NFE budget, CFO already outperforms AR; increasing the

budget to 200% yields further gains and stable high accuracy, while additional NFE up to 400% brings diminishing returns with only negligible improvements. To connect NFE to actual runtime, we also report wall-clock inference time and accuracy for AR and CFO+RK4 on identical hardware with the same spatial backbone in Table 10 and Figure 8. It confirms that the theoretical reduction in NFE translates directly to wall-clock time savings.

Table 2: Influence of NFE on Quintic CFO (RK4 solver) accuracy (Relative $L_2$ Error) at the final time step, compared to the autoregressive baseline.

| NFE (% AR) | Lorenz | Burgers | DR | SWE |
|---|---|---|---|---|
| AR (100%) | 0.1481 | 0.0647 | 0.3850 | 0.1048 |
| 50% | 0.0919 | 0.0090 | 0.0876 | 0.0698 |
| 100% | 0.0655 | 0.0089 | 0.0718 | 0.0154 |
| 200% | 0.0674 | 0.0090 | 0.0608 | 0.0060 |
| 400% | 0.0645 | 0.0089 | 0.0593 | 0.0061 |

**Impact of Noise Schedule.** We investigate the impact of the noise schedule on model performance. The noise schedule $\gamma(t) = \gamma_0 \frac{(t-t_k)^m (t_{k+1}-t)^m}{(t_{k+1}-t_k)^{2m}}$ for $t \in [t_k, t_{k+1}]$, where $m$ controls smoothness and $\gamma_0$ controls magnitude. Figure 7 shows the impact of noise magnitude $\gamma_0$ on the SWE and DR benchmarks. Performance is stable for small noise levels ($\gamma_0 \in [0, 10^{-4}]$) but degrades as the noise magnitude increases. For our experiments, we set $m = 3$ to ensure the noise schedule is $C^2$ continuous, matching the smoothness of the quintic spline.

**Comparison with Other Baselines.** Besides AR, we also compare CFO with two other baselines. The first is continuous-time neural ODE model (Chen et al., 2018) trained with a teacher-forcing strategy. CFO attains higher accuracy with substantially less training time (table 3), and the continuous-time supervision of CFO leads to more stable optimization (Figure 9). Second, we compare against PDE-Refiner (Lippe et al., 2023), a strong baseline for long-horizon rollouts. CFO outperforms PDE-Refiner on long-horizon predictions while using fewer computational resources, since PDE-Refiner requires diffusion-style denoising simulations during training.

**Reverse-time Inference.** We study CFO's performance of backward integration on the dissipative Burgers' equation. For Figure 10(a), we fix a terminal time $t_\star = 1$ and integrate the flow backward over varying backward horizons within $[0, t_\star)$. The panel reports how the error accumulates with increasing backward horizon under different noise levels added to the terminal state. Figure 10(b) visualizes initial-condition recovery by integrating backward from different terminal times to $t = 0$. Despite the ill-posed nature of inverse problems for dissipative PDEs, CFO yields reasonable reverse-time reconstructions over short backward horizons before errors grow.

**Temporal Extrapolation Beyond the Training Horizon.** To assess temporal extrapolation, we train CFO only on the first half of each trajectory $[0, T/2]$, and evaluate rollouts over the full horizon $[0, T]$ (Table 11). The error accumulation curves in Figure 11 show that, for Lorenz, Burgers, and DR, the extrapolation errors remains nearly identical to full-horizon training results, indicating that CFO has learned the underlying dynamics rather than memorizing the trajectories. For SWE, the error increases to $2.28 \times 10^{-2}$, but this still remains approximately $4\times$ lower than the full-horizon training autoregressive baseline.

**Neural Network Architectures.** We assess the effect of the backbone on CFO by swapping the spatial operator while keeping the training recipe fixed. For 1D Burgers we use an FNO (Li et al., 2020a); for 2D diffusion–reaction and 2D shallow water (SWE) we use the DiT (Peebles & Xie, 2023) Architecture. Results in Table 12 show comparable accuracy for backbones, with only modest variations. This indicates that CFO is largely architecture-agnostic: temporal continuity is handled by CFO, while the spatial inductive bias can be chosen to suit resolution, memory, or hardware constraints.

Table 3: Comparison of CFO with other baselines: Neural ODE (Chen et al., 2018) and PDE-Refiner (Lippe et al., 2023).

| Equation | Method | Rel. $L_2$ Error | Training Time (s/batch) |
|---|---|---|---|
| Lorenz | Neural ODE | 0.101 | 0.133 |
| | **CFO** | **0.0453** | **0.00350** |
| Burgers | Neural ODE | 0.0275 | 3.38 |
| | **CFO** | **0.00589** | **0.00920** |
| DR | PDE-Refiner | 0.125 | 1.38 |
| | **CFO** | **0.044** | **0.40** |
| SWE | PDE-Refiner | 0.093 | 1.40 |
| | **CFO** | **0.005** | **0.40** |

## 6 CONCLUSIONS AND FUTURE WORK

We introduced CFO, a continuous-time neural operator that learns PDE dynamics by matching the analytic velocity of spline-based interpolants. By repurposing flow matching to avoid ODE solver backpropagation during training, CFO achieves a unique combination of capabilities: training on irregular, trajectory-specific time grids; inference at arbitrary temporal resolutions; and accurate predictions from severely subsampled data. Across four benchmarks, models trained on only 25% of irregularly sampled time points outperform fully-trained autoregressive baselines, demonstrating that physics-aware probability path design can overcome data scarcity.

Several directions warrant future investigation. *Spatial resolution*: While CFO handles temporal irregularity, coupling it with mesh-agnostic operators would enable full spatio-temporal resolution invariance. *Inference acceleration*: Though CFO matches autoregressive efficiency with modest function evaluations, distilling the learned flow into consistency models (Song et al., 2023) could enable single-step predictions. *Physics integration*: When governing equations are partially known, augmenting training with physics-informed constraints (Huang et al., 2024) could improve backward integration stability. *Adaptive splines*: Current fixed-order splines face a trade-off between computational cost and accuracy; learned splines with curvature regularization (Luo et al., 2025) could adapt smoothness to local dynamics.

CFO demonstrates that continuous-time modeling need not sacrifice computational practicality. By bridging the efficiency of discrete methods with the flexibility of continuous formulations, it opens new possibilities for learning from real-world data with irregular temporal measurements, a critical step toward practical neural PDE solvers.

## REPRODUCIBILITY STATEMENT

To facilitate reproducibility, we provide extensive implementation details, ablation studies, and theoretical clarifications throughout the main paper and appendix. Section 4 describes the proposed model architecture and training procedure, while Section 5 outlines the experimental setup across all benchmark tasks. The theoretical results are included in Appendices A.1–A.2, and the full algorithms are summarized in Appendix A.3. All hyperparameters and data preprocessing steps are detailed in Appendix A.4. Code is available at `https://github.com/shannon-hou/CFO_official`.

## ACKNOWLEDGMENTS

This work was supported by the US Department of Energy under the Advanced Scientific Computing Research program (grant DE-SC0024563), and the Penn AI Fellowship program.

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

# A APPENDIX

## A.1 SPLINE CONSTRUCTION

This section details the two steps of our spline construction: (i) estimating derivatives at the knots and (ii) piecewise interpolation between knots. We provide a general finite-difference framework for derivative estimation on nonuniform grids, and we give the closed-form quintic Hermite interpolant on each interval.

The results follow standard numerical references; see, e.g., (Strikwerda, 2004; Hildebrand, 1987).

### A.1.1 FINITE DIFFERENCE STENCIL ON A NONUNIFORM GRID

Let $f \in C^{m+1}(I)$ on an interval $I \subset \mathbb{R}$ and let $x_0, x_1, \ldots, x_m \in I$ be pairwise distinct with point of interest $x_0$. Set $\Delta_j := x_j - x_0$ and $h := \max_{0 \le j \le m} |\Delta_j|$. Fix $k \in \{0, 1, \ldots, m\}$.

Define the Vandermonde matrix

$$P := \big[ P_{rj} \big]_{r,j=0}^m, \qquad P_{rj} = (x_j - x_0)^r = \Delta_j^r.$$

Let $e_k \in \mathbb{R}^{m+1}$ denote the $k$-th standard basis vector. We choose weights $w = (w_0, \ldots, w_m)^\top$ by solving

$$P w = k! \, e_k,$$

Then the $(m+1)$-point finite-difference approximation of $f^{(k)}(x_0)$ is

$$D^{(k)} f(x_0) := \sum_{j=0}^m w_j \, f(x_j),$$

and the error is controlled by

$$f^{(k)}(x_0) - D^{(k)} f(x_0) = \mathcal{O}(h^{m+1-k}).$$

### A.1.2 QUINTIC HERMITE INTERPOLATION

Suppose $f \in C^6([t, t'])$ the value, first-, and second-derivatives are $s, d, a$ and $s', d', a'$ at endpoints respectively. Set $h := t' - t$ and $\tau := \dfrac{x - t}{h} \in [0, 1]$. The unique quintic Hermite interpolant $H(x)$ that matches values, first-, and second-derivatives at both endpoints can be written analytically on $[t, t']$ as

$$
\begin{aligned}
H(x) = {}& s \, H_{00}(\tau) + (d \, h) \, H_{10}(\tau) + (a \, h^2) \, H_{20}(\tau) \\
& + s' \, H_{01}(\tau) + (d' \, h) \, H_{11}(\tau) + (a' \, h^2) \, H_{21}(\tau),
\end{aligned}
$$

where the quintic Hermite basis polynomials are

$$
\begin{aligned}
H_{00}(\tau) &= 1 - 10\tau^3 + 15\tau^4 - 6\tau^5, & H_{10}(\tau) &= \tau - 6\tau^3 + 8\tau^4 - 3\tau^5, \\
H_{20}(\tau) &= \tfrac{1}{2}\big(\tau^2 - 3\tau^3 + 3\tau^4 - \tau^5\big), & H_{01}(\tau) &= 10\tau^3 - 15\tau^4 + 6\tau^5, \\
H_{11}(\tau) &= -4\tau^3 + 7\tau^4 - 3\tau^5, & H_{21}(\tau) &= \tfrac{1}{2}\big(\tau^3 - 2\tau^4 + \tau^5\big).
\end{aligned}
$$

These satisfy the endpoint conditions

$$H_{00}(0) = 1, \; H'_{10}(0) = 1, \; H''_{20}(0) = 1, \quad H_{01}(1) = 1, \; H'_{11}(1) = 1, \; H''_{21}(1) = 1,$$

and all other values/derivatives up to order 2 vanish at the opposite endpoint.

### A.1.3 SPLINE SELECTION

**Spline fitting cost.** The cost of spline fitting is negligible. The finite-difference approximation and spline construction involve only simple algebraic operations (no iterative solvers). They are performed once per trajectory as a preprocessing step and reused throughout training. The computational complexity is $\mathcal{O}(\text{spline order} \times \# \text{ time segments} \times \# \text{ spatial grid points})$ per trajectory, which is negligible compared to neural operator backpropagation.

**Practical guidelines for choosing splines.**

1. Given the low cost of spline fitting, we use quintic rather than linear splines as the default, since quintic splines capture acceleration (second derivatives), which is central to many physical laws (e.g., Newton's second law, wave equations). At the same time, we do not go beyond quintic because higher orders bring diminishing returns in accuracy.

2. Linear splines are adequate when (i) time sampling is very dense (e.g., 100% keep ratio), or (ii) the underlying dynamics are known to be non-smooth (e.g., shocks or discontinuities), where enforcing higher-order global smoothness may be undesirable.

3. Adaptive spline selection is an exciting direction for future work. A dynamic approach could involve learning the optimal spline parameters jointly with the operator, potentially incorporating physics-informed priors to adapt the smoothness.

## A.2 PROBABILITY-FLOW TRANSPORT

This section provides the theoretical guarantee of CFO's probability-flow transport properties.

**Assumption A.1.** *Let $\mathbf{u} = (u(t_0), \ldots, u(t_N)) \sim q_{\mathcal{T}}$ with $0 = t_0 < \cdots < t_N = 1$. Assume:*

1. *(**Knot densities**) For each $k = 0, \ldots, N$, $u(t_k)$ admits a strictly positive Lebesgue density $q_k : \mathbb{R}^d \to (0, \infty)$.*

2. *(**Spline in time**) The map $t \mapsto s(t; \mathbf{u})$ is $C^1$ on $[0, 1]$, satisfies $s(t_k; \mathbf{u}) = u(t_k)$, and $\partial_t s$ is dominated in $L^1$: there exists an integrable random variable $G(\mathbf{u}) \in L^1$ such that*

$$\|\partial_t s(t; \mathbf{u})\| \leq G(\mathbf{u}) \quad \text{for all } t \in [0, 1], \qquad \mathbb{E}_{\mathbf{u}}[G(\mathbf{u})] < \infty.$$

*where $\|\cdot\|$ denotes the Euclidean norm on $\mathbb{R}^d$.*

3. *(**Noise schedule**) $\gamma \in C^1([0, 1])$ with $\gamma(t_k) = 0$ for all $k$, $\gamma(t) > 0$ for $t \notin \mathcal{T}$, and $\|\gamma'\|_\infty < \infty$.*

4. *(**Independent Gaussian**) $Z \sim \mathcal{N}(0, I_d)$ independent of $\mathbf{u}$.*

**Proposition A.1.** *Under Assumption A.1, define*

$$I(t; \mathbf{u}) = s(t; \mathbf{u}) + \gamma(t)Z, \qquad \partial_t I(t; \mathbf{u}) = \partial_t s(t; \mathbf{u}) + \gamma'(t)Z.$$

*Then:*

1. *For every $t \in [0, 1]$, $I(t; \mathbf{u})$ admits a strictly positive Lebesgue density $\rho(t, \cdot)$. Moreover, for every knot $t_k$, $\rho(t_k, x) = q_k(x)$ for $x \in \mathbb{R}^d$.*

2. *Let*
$$v^\star(t, x) := \mathbb{E}[\partial_t I(t; \mathbf{u}) \mid I(t; \mathbf{u}) = x].$$
*Then for every test function $\psi \in C_c^\infty((0, 1) \times \mathbb{R}^d)$,*

$$\int_0^1 \int_{\mathbb{R}^d} \left(\partial_t \psi(t, x) + \nabla \psi(t, x) \cdot v^\star(t, x)\right) \rho(t, x) \, dx \, dt = 0.$$

*Equivalently, the continuity equation $\partial_t \rho + \nabla \cdot (\rho v^\star) = 0$ holds in $\mathcal{D}'((0, 1) \times \mathbb{R}^d)$.*

*Proof.* (1) Fix $t \notin \mathcal{T}$. From the independence of $\mathbf{u}$ and $Z$, the law of

$$I(t; \mathbf{u}) = s(t; \mathbf{u}) + \gamma(t)Z$$

is the convolution of $\mu_t := \text{Law}(s(t; \mathbf{u}))$ with the non-degenerate Gaussian $\mathcal{N}(0, \gamma^2(t)I_d)$. Hence $I(t; \mathbf{u})$ admits the Lebesgue density

$$\rho(t, \cdot) = \varphi_{\gamma(t)} * \mu_t.$$

Because $\varphi_{\gamma(t)}$ is a strictly positive $C^\infty$ kernel, convolution with $\varphi_{\gamma(t)}$ regularizes $\mu_t$: it follows that $\rho(t, \cdot) \in C^\infty(\mathbb{R}^d)$ and $\rho(t, x) > 0$ for all $x \in \mathbb{R}^d$.

At a knot $t = t_k$, $\gamma(t_k) = 0$ and $s(t_k; \mathbf{u}) = u(t_k)$, hence

$$I(t_k; \mathbf{u}) = u(t_k)$$

By Assumption A.1(1), $u(t_k)$ has strictly positive Lebesgue density $q_k$; therefore $\rho(t_k, \cdot) = q_k(\cdot)$.

(2) Take test function $\psi \in C_c^\infty((0, 1) \times \mathbb{R}^d)$ and define

$$F(t) := \mathbb{E}[\psi(t, I(t; \mathbf{u}))], \qquad t \in (0, 1).$$

Since $\psi$ has compact support in $t$, $F(t) = 0$ near $t = 0$ and $t = 1$, and thus

$$\int_0^1 F'(t)\, dt = F(1) - F(0) = 0,$$

once $F'$ exists and is integrable.

Since the map $t \mapsto I(t; \mathbf{u})$ is $C^1$, by the chain rule,

$$\frac{d}{dt}\psi(t, I(t; \mathbf{u})) = \partial_t \psi(t, I(t; \mathbf{u})) + \nabla\psi(t, I(t; \mathbf{u})) \cdot \partial_t I(t; \mathbf{u}).$$

Since $\psi \in C_c^\infty((0, 1) \times \mathbb{R}^d)$, $\partial_t \psi$ and $\nabla\psi$ are bounded.

Moreover, by Assumption A.1(2), there exists $G(\mathbf{u}) \in L^1$ such that $\|\partial_t s(t; \mathbf{u})\| \leq G(\mathbf{u})$ for all $t \in [0, 1]$ Hence, for all $t \in [0, 1]$,

$$\|\partial_t I(t; \mathbf{u})\| \leq \|\partial_t s(t; \mathbf{u})\| + |\gamma'(t)|\, \|Z\| \leq G(\mathbf{u}) + \|\gamma'\|_\infty \|Z\| =: G_I(\mathbf{u}, Z).$$

The random variable $G_I$ is integrable under the joint law of $(\mathbf{u}, Z)$ since

$$\mathbb{E}_{\mathbf{u}, Z}[G_I] = \mathbb{E}_\mathbf{u}[G(\mathbf{u})] + \|\gamma'\|_\infty \mathbb{E}_Z \|Z\| < \infty.$$

Therefore,

$$\left| \partial_t \psi(t, I(t; \mathbf{u})) + \nabla\psi(t, I(t; \mathbf{u})) \cdot \partial_t I(t; \mathbf{u}) \right| \leq \|\partial_t \psi\|_\infty + \|\nabla\psi\|_\infty\, G_I(\mathbf{u}, Z),$$

and the right-hand side is an $L^1(\mathbf{u}, Z)$ dominating random variable independent of $t$. Thus, dominated convergence justifies differentiating under the expectation, yielding

$$F'(t) = \mathbb{E}[\partial_t \psi(t, I(t; \mathbf{u})) + \nabla\psi(t, I(t; \mathbf{u})) \cdot \partial_t I(t; \mathbf{u})].$$

Next, by the tower property conditioning on $I(t; \mathbf{u})$, we have

$$\mathbb{E}\big[\nabla\psi(t, I(t; \mathbf{u})) \cdot \partial_t I(t; \mathbf{u})\big] = \mathbb{E}\big[\nabla\psi(t, I(t; \mathbf{u})) \cdot v^\star(t, I(t; \mathbf{u}))\big].$$

Since $I(t; \mathbf{u})$ has density $\rho(t, \cdot)$ for $t \in (0, 1)$, we obtain for $t \in (0, 1)$,

$$F'(t) = \int_{\mathbb{R}^d} \Big( \partial_t \psi(t, x) + \nabla\psi(t, x) \cdot v^\star(t, x) \Big) \rho(t, x)\, dx.$$

Integrating over $t \in (0, 1)$ and using $\int_0^1 F'(t)\, dt = 0$ yields

$$\int_0^1 \int_{\mathbb{R}^d} \Big( \partial_t \psi(t, x) + \nabla\psi(t, x) \cdot v^\star(t, x) \Big) \rho(t, x)\, dx\, dt = 0,$$

which is exactly the weak form of $\partial_t \rho + \nabla \cdot (\rho v^\star) = 0$ on $(0, 1) \times \mathbb{R}^d$. $\qquad\square$

**Proposition A.2.** *The conditional mean velocity $v^\star(t, x) := \mathbb{E}[\partial_t I(t; \mathbf{u}) \mid I(t; \mathbf{u}) = x]$ is the unique minimizer of*

$$\mathcal{L}(v) = \int_0^1 \mathbb{E}\Big[ \big\| v(t, I(t; \mathbf{u})) - \partial_t I(t; \mathbf{u}) \big\|^2 \Big] dt, \tag{9}$$

*Proof.*

$$\mathbb{E}\left[\left\|v(t, I(t; \mathbf{u})) - \partial_t I(t; \mathbf{u})\right\|^2\right] = \mathbb{E}\left[\left\|v(t, I(t; \mathbf{u})) - \mathbb{E}[\partial_t I(t; \mathbf{u}) \mid I(t; \mathbf{u})]\right\|^2\right]$$
$$+ \mathbb{E}\left[\left\|\mathbb{E}[\partial_t I(t; \mathbf{u}) \mid I(t; \mathbf{u})] - \partial_t I(t; \mathbf{u})\right\|^2\right]$$
$$\geq \mathbb{E}\left[\left\|\mathbb{E}[\partial_t I(t; \mathbf{u}) \mid I(t; \mathbf{u})] - \partial_t I(t; \mathbf{u})\right\|^2\right].$$

The equality holds if and only if $v(t, x) = \mathbb{E}[\partial_t I(t; \mathbf{u}) \mid I(t; \mathbf{u}) = x]$. $\qquad\square$

**Remark A.1.** *By Proposition A.1, the marginals $\rho(t, \cdot)$ solve the continuity equation with velocity $v^\star$. Under standard regularity conditions ensuring existence/uniqueness of the flow for $\dot{u}(t) = v^\star(t, u(t))$ (e.g. local Lipschitz in $x$), this ODE transports $q_0$ to $\rho(t, \cdot)$, and in particular matches the knot marginals $q_k$ at $t = t_k$.*

*By Proposition A.2, $v^\star$ is the minimizer of the regression objective (9). Therefore, if the learned operator $\mathcal{N}_\theta$ matches $v^\star$ (e.g., at the minimizer of (9)), then the forward ODE 8 induces the same probability-flow transport across the knots.*

## A.3 ALGORITHM

We summarize the numerical procedures used in the main text. Algorithm 1 describes the construction of quintic splines from discrete snapshots. Algorithm 2 gives a single-sample CFO training update based on these splines; in practice, we sample trajectories and use minibatches. Algorithm 3 outlines the corresponding inference procedure.

---

**Algorithm 1** Quintic Spline Construction

---

1: **Input:** Snapshots $\mathbf{u} = \left(u(t_0), \ldots, u(t_N)\right)$ with time grid $\mathcal{T} = (t_i)_{i=0}^N$.

2: **for** $i = 0, \ldots, N$ **do**

3:     Estimate first and second time derivatives $d_i \approx \partial_t u(t_i)$ and $a_i \approx \partial_{tt} u(t_i)$ using the finite-difference stencils in Appendix A.1.1; for the three-point scheme, use the explicit expression in Eq. (5).

4: **end for**

5: **for** $i = 0, \ldots, N-1$ **do**

6:     Define the local coordinate $\tau = \dfrac{t - t_i}{t_{i+1} - t_i} \in [0, 1]$.

7:     On $[t_i, t_{i+1}]$, construct $s(t)$ via the quintic Hermite form in Appendix A.1.2 using the knot data $\left(u(t_i), u(t_{i+1}), d_i, d_{i+1}, a_i, a_{i+1}\right)$.

8: **end for**

9: **Output:** $s(t)$ (the collection of per-segment coefficients).

---

**Algorithm 2** CFO Training (One-Sample Example)

---

1: **Input:** A trajectory $\mathbf{u}$ with time grid $\mathcal{T}$, neural operator $\mathcal{N}_\theta(t, u)$, noise schedule $\gamma(t)$.

2: Construct spline $s(t)$ with $\left(\mathbf{u}, \mathcal{T}\right)$; for the quintic spline, use Algorithm 1.

3: **for** iter $= 1, \ldots, S$ **do**

4:     Sample $t \sim \text{Unif}[0, 1]$ and $z \sim \mathcal{N}(0, I)$.

5:     $x \leftarrow s(t) + \gamma(t)\, z$.

6:     $v^{\text{target}} \leftarrow \partial_t s(t) + \gamma'(t)\, z$.

7:     Update $\theta$ using loss $\mathcal{L}(\theta) = \|\mathcal{N}_\theta(t, x) - v^{\text{target}}\|_2^2$.

8: **end for**

9: **Output:** $\mathcal{N}_\theta$

---

---

**Algorithm 3** CFO Inference

---

1: **Input:** Trained neural operator $\mathcal{N}_\theta(t, u)$, initial time $t_0$ and state $u_0$, evaluation times $\mathcal{T}_{\text{eval}} = (t_k)_{k=0}^K$ with $t_0 < t_1 < \cdots < t_K$, ODE solver (e.g., RK4).

2: $\hat{u}(t_0) \leftarrow u_0$.

3: **for** $k = 0, \ldots, K - 1$ **do**

4: $\quad \hat{u}(t_{k+1}) \leftarrow \hat{u}(t_k) + \int_{t_k}^{t_{k+1}} \mathcal{N}_\theta(t, \hat{u}(t)) \, dt$ (approximated by the chosen ODE solver).

5: **end for**

6: **Output:** $\{\hat{u}(t_k)\}_{k=0}^K$

---

### A.4 EXPERIMENTAL DETAILS

#### A.4.1 METRICS

We evaluate model accuracy using the relative $L_2$ norm between the predicted $_{\text{pred}}$ and ground-truth $u_{\text{true}}$ trajectories, averaged over all test trajectories:

$$\text{Relative } L_2 = \frac{1}{N_{\text{test}}} \sum_{k=1}^{N_{\text{test}}} \frac{\left( \sum_{j=1}^{N_t} \sum_{i=1}^{N_x} \left\| u_{\text{pred}}^{(k)}(t_j, x_i) - u_{\text{true}}^{(k)}(t_j, x_i) \right\|^2 \right)^{1/2}}{\left( \sum_{j=1}^{N_t} \sum_{i=1}^{N_x} \left\| u_{\text{true}}^{(k)}(t_j, x_i) \right\|^2 \right)^{1/2}}.$$

#### A.4.2 NEURAL NETWORK ARCHITECTURES

We evaluate Multi-Layer Perceptrons (MLPs), U-Nets (Ronneberger et al., 2015), Diffusion Transformers (DiTs) (Peebles & Xie, 2023), and Fourier Neural Operators (FNOs) (Li et al., 2023). CFO models receive sinusoidal time embeddings; autoregressive baselines do not.

**MLP** A 5-layer MLP with hidden dims [128, 256, 256, 256, 128] and ReLU activations; time-embedding dimension 16 ($\approx 0.202$M parameters).

**FNO** A 1D Fourier Neural Operator with width 128, 5 spectral blocks, and $m = 6$ retained Fourier modes. Each block applies an rFFT, learned complex-mode multiplication with a $1 \times 1$ residual conv, and GELU. Inputs are augmented with a normalized grid and lifted to width 128; a two-layer head maps to one output channel ($\approx 0.591$M parameters).

**1D U-Net** A minimal 1D U-Net with two scales (channel_mult $= [64, 128]$) using Conv–GroupNorm–Swish blocks, $4 \times 1$ stride-2 downsampling, transposed-convolution upsampling, and a $1 \times 1$ head; time-embedding dimension 64 ($\approx 0.602$M parameters).

**2D U-Net** A 2D U-Net with three scales (channel_mult $= [64, 128, 256]$), Conv–GroupNorm–Swish blocks, $4 \times 4$ stride-2 downsampling, transposed-convolution upsampling, and a $1 \times 1$ head; time-embedding dimension 256 ($\approx 8.1$M parameters).

**DiT** A DiT with $8 \times 8$ patch embedding, hidden size 384, depth 4, 8 attention heads, and MLP ratio 6. Fixed 2D sinusoidal positional embeddings are added to patch tokens; a linear head reconstructs one output channel. ($\approx 13.4$M parameters).

#### A.4.3 TEACHER-FORCING TRAINING FOR BASELINE MODELS

We summarize the teacher-forcing schemes used to train the autoregressive and neural ODE baselines.

**Autoregressive baselines.** Let $\{u(t_i)\}_{i=0}^N$ denote a trajectory sampled on a uniform time grid. The AR model $F_\phi$ is trained as a one-step predictor $\hat{u}(t_{i+1}) = F_\phi(u(t_i))$. At each step the input is the

ground-truth state $u(t_i)$ rather than the model prediction (teacher forcing). The training loss is the average one-step error

$$\mathcal{L}_{\mathrm{AR}}(\phi) = \frac{1}{N} \sum_{i=0}^{N-1} \left\| F_\phi\big(u(t_i)\big) - u(t_{i+1}) \right\|_2^2.$$

At inference time, we roll out autoregressively by feeding the model's own prediction back as input from ground-truth initial condition $\hat{u}(t_0) = u(t_0)$:

$$\hat{u}(t_{i+1}) = F_\phi(\hat{u}(t_i)), \quad i = 0, \cdots, N-1.$$

**Neural ODE baseline.** For the continuous-time baseline, we parameterize the right-hand side as a neural network $f_\psi(t, u)$ and define the ODE

$$\frac{d}{dt}\tilde{u}(t) = f_\psi\big(t, \tilde{u}(t)\big).$$

We train this model with a teacher-forcing strategy on short time intervals. For each pair $(t_i, t_{i+1})$ in a trajectory, we initialize the ODE solver at the ground-truth state $\tilde{u}(t_i) = u(t_i)$ and numerically integrate to obtain $\tilde{u}(t_{i+1}) = u(t_i) + \int_{t_i}^{t_{i+1}} f_\psi(\tau, \tilde{u}(\tau))d\tau$. The loss is the average mismatch between the integrated solution and the next ground-truth snapshot:

$$\mathcal{L}_{\mathrm{NODE}}(\psi) = \frac{1}{N} \sum_{i=0}^{N-1} \left\| \tilde{u}(t_{i+1}) - u(t_{i+1}) \right\|_2^2.$$

This setup uses ground-truth states as starting points for each integration interval.

### A.4.4 LORENZ SYSTEM

We consider the chaotic ODE system with spatial-derivative-free dynamics:

$$\frac{dx}{dt} = \sigma(y - x), \quad \frac{dy}{dt} = x(\rho - z) - y, \quad \frac{dz}{dt} = xy - \beta z, \qquad t \in [0, 5],$$

with $(\sigma, \rho, \beta) = (10, 28, 8/3)$.

**Dataset.** The initial condition $(x_0, y_0, z_0)$ is sampled uniformly from the box $[-5, 5]^3$. The dataset comprises 9000 training, 500 validation, and 500 testing trajectories. Each trajectory is sampled at 1001 uniformly spaced time points over a horizon of $5.0s$.

For the irregular sampling experiments, we create subsampled datasets by randomly keeping 50% or 25% of the time points for each trajectory, with the selection performed independently for each trajectory. This procedure is consistent across all equations.

**Neural Solver Choice.** Both the CFO and the autoregressive baseline use the same MLP architecture. For CFO models, normalized time $t \in [0, 1]$ is encoded using a 16-band sinusoidal embedding.

**Training of CFO and AR.** The model is trained for 200,000 steps using the Adam optimizer with a batch size of 2048, a learning rate of $10^{-4}$, and betas $(\beta_1, \beta_2) = (0.9, 0.99)$, and evaluated every 5000 steps. The noise schedule parameter is set to $\gamma_0 = 10^{-5}$.

**Training of Neural ODE.** We use the Tsit5 adaptive Runge–Kutta solver (Tsitouras, 2011) with absolute tolerance $10^{-6}$ and relative tolerance $10^{-4}$. Backpropagation through the ODE solution is performed using Diffrax's `RecursiveCheckpointAdjoint` method (Kidger, 2021) for memory-efficient gradients. The other training setting matches CFO and AR.

### A.4.5 BURGERS' EQUATION

Consider the 1D Burgers' equation benchmark investigated in (Wang et al., 2021):

$$\frac{du}{dt} = \nu \frac{d^2 u}{dx^2} - u \frac{du}{dx}, \quad (x, t) \in (0, 1) \times (0, 1],$$
$$u(x, 0) = u_0(x), \qquad x \in (0, 1),$$

with the viscosity $\nu = 0.01$.

Table 4: Endpoint error and empirical order $p$ for the spline-implied velocity on the Lorenz system. Errors are average relative $L_2$ at trajectory segment endpoints ($\tau \in \{0,1\}$) over 50000 segments. $\Delta t = 0.005$ s.

| Spline type | Step size | Error ($\tau{=}0$) | $p$ ($\tau{=}0$) | Error ($\tau{=}1$) | $p$ ($\tau{=}1$) |
|---|---|---|---|---|---|
| Quintic | $\Delta t$ | $5.33 \times 10^{-4}$ | - | $5.32 \times 10^{-4}$ | - |
| | $2\Delta t$ | $2.16 \times 10^{-3}$ | 2.02 | $2.16 \times 10^{-3}$ | 2.02 |
| | $4\Delta t$ | $8.63 \times 10^{-3}$ | 2.00 | $8.63 \times 10^{-3}$ | 2.00 |
| Linear | $\Delta t$ | $2.49 \times 10^{-2}$ | - | $2.48 \times 10^{-2}$ | - |
| | $2\Delta t$ | $4.97 \times 10^{-2}$ | 1.00 | $4.96 \times 10^{-2}$ | 1.00 |
| | $4\Delta t$ | $9.92 \times 10^{-2}$ | 1.00 | $9.88 \times 10^{-2}$ | 1.00 |

**Dataset.**  The initial condition $u_0(x)$ is generated from a Gaussian Random Field (GRF) $\sim \mathcal{N}\left(0, 25^2(-\Delta + 5^2 I)^{-4}\right)$ with periodic boundary conditions. The dataset consists of 1000 training, 100 validation, and 100 testing trajectories. Each trajectory is sampled at 101 uniformly spaced time points over $[0, 1s]$. The spatial domain $[0, 1]$ is discretized into 100 points.

**Neural Solver Choice.**  For CFO, we use the 1D U-Net; results appear in Table 1. To demonstrate flexibility, our ablation study part 5.2 include a FNO variant. The autoregressive baseline mirrors these architectures (MLP, U-Net, FNO) with time embeddings disabled; we report only the best-performing variant—MLP—in Table 1.

Table 5: Relative $L_2$ Error of autoregressive baselines with different architectures on the 1D Burgers' equation. The MLP architecture, as the best-performing variant, is used for comparison in Table 1.

| Architecture | MLP | U-Net | FNO |
|---|---|---|---|
| Relative $L_2$ Error | $3.34 \times 10^{-2}$ | 2.56 | 1.16 |

**Training of CFO and AR.**  The model is trained for 60,000 steps using the Adam optimizer with a batch size of 256, a learning rate of $10^{-4}$, and betas $(\beta_1, \beta_2) = (0.9, 0.99)$. The model is evaluated every 5000 steps. The noise schedule parameter is set to $\gamma_0 = 10^{-5}$.

**Training of Neural ODE.**  We use the Tsit5 adaptive Runge–Kutta solver (Tsitouras, 2011) with absolute tolerance $10^{-6}$ and relative tolerance $10^{-4}$. Backpropagation through the ODE solution is performed using Diffrax's `RecursiveCheckpointAdjoint` method (Kidger, 2021) for memory-efficient gradients. The other training setting matches CFO and AR.

### A.4.6   DIFFUSION-REACTION EQUATION

We consider the 2D diffusion-reaction equation from PDEBench (Takamoto et al., 2022):

$$\partial_t u = D_u(\partial_{xx} u + \partial_{yy} u) + R_u(u, v),$$
$$\partial_t v = D_v(\partial_{xx} v + \partial_{yy} v) + R_v(u, v),$$

where $u = u(t, x, y)$ is the activator and $v = v(t, x, y)$ denotes the inhibitor on the domain $(t, x, y) \in (0, 5] \times (-1, 1)^2$. The reaction terms are given by

$$R_u(u, v) = u - u^3 - k - v,$$
$$R_v(u, v) = u - v,$$

with $k = 5 \times 10^{-3}$. The diffusion coefficients are $D_u = 1 \times 10^{-3}$ and $D_v = 5 \times 10^{-3}$.

**Dataset.**  We directly use the dataset provided in PDEBench (Takamoto et al., 2022). The initial condition is generated as standard normal random noise $u(0, x, y) \sim \mathcal{N}(0, 1.0)$. The dataset consists of 900 training, 50 validation, and 50 testing trajectories of activator and inhibitor (2 channels). Each trajectory is sampled at 101 uniformly spaced time points over $[0, 5s]$, but the first time step is truncated to ensure stability. The spatial domain is discretized into a $128 \times 128$ grid.

**Neural Solver Choice.** For CFO, we use the 2D U-Net described above; results shown in Table 1. To demonstrate flexibility, our ablation study part 5.2 include a DiT variant. The autoregressive baseline mirrors these architectures (U-Net, DiT) with time embeddings disabled; we report only the best-performing variant—U-Net—in Table 1.

Table 6: Relative $L_2$ Error of autoregressive baselines with different architectures on the 2D diffusion-reaction Equation. The U-Net architecture, as the best-performing variant, is used for comparison in Table 1.

| Architecture | U-Net | DiT |
|---|---|---|
| Relative $L_2$ Error | $(4.23 \pm 0.59) \times 10^{-1}$ | $(9.21 \pm 0.10) \times 10^{-1}$ |

**Training of CFO and AR.** The model is trained for 5000 steps using the Adam optimizer with a batch size of 256, a learning rate of $2 \times 10^{-4}$, and betas $(\beta_1, \beta_2) = (0.9, 0.99)$. The model is evaluated every 500 steps. The noise schedule parameter is set to $\gamma_0 = 10^{-5}$.

**Training of PDE-Refiner.** For the PDE-Refiner baseline, we follow the training setup from the original paper (Lippe et al., 2023) for 2D equations and use the 2D U-Net architecture with a parameter count and number of training steps matched to CFO for a fair comparison.

A.4.7 SHALLOW WATER EQUATION

We consider the 2D shallow water equations from PDEBench (Takamoto et al., 2022):

$$\partial_t h + \partial_x(hu) + \partial_y(hv) = 0,$$

$$\partial_t(hu) + \partial_x \left( u^2 h + \frac{1}{2} g_r h^2 \right) + \partial_y(uvh) = -g_r h \partial_x b,$$

$$\partial_t(hv) + \partial_y \left( v^2 h + \frac{1}{2} g_r h^2 \right) + \partial_x(uvh) = -g_r h \partial_y b,$$

where $h$ is the water depth, $(u, v)$ are the velocities, $b$ is the spatially varying bathymetry, and $g_r$ is the gravitational acceleration.

**Dataset.** We use the dataset from PDEBench (Takamoto et al., 2022). The initial condition for water depth is generated from a GRF, with initial velocities set to zero. The dataset consists of 900 training, 50 validation, and 50 testing trajectories of water depth (one channel). Each trajectory is sampled at 101 uniformly spaced time points on [0,1], with the first time step truncated. The spatial domain is a $128 \times 128$ grid.

**Neural Solver Choice.** For CFO, we use the 2D U-Net; results are shown in Table 1. To demonstrate flexibility, our ablation study in Section 5.2 includes a DiT variant. The autoregressive baseline mirrors these architectures (U-Net, DiT) with time embeddings disabled; we report only the best-performing variant (U-Net) in Table 1 for comparison.

Table 7: Relative $L_2$ Error of autoregressive baselines with different architectures on the 2D Shallow Water Equation. The DiT architecture, as the best-performing variant, is used for comparison in Table 1.

| Architecture | U-Net | DiT |
|---|---|---|
| Relative $L_2$ Error | $3.16 \times 10^{-1}$ | $9.04 \times 10^{-2}$ |

**Training of CFO and AR.** The model is trained for 5000 steps using the Adam optimizer with a batch size of 256, a learning rate of $10^{-5}$, and betas $(\beta_1, \beta_2) = (0.9, 0.99)$. The model is evaluated every 500 steps. The noise schedule parameter is set to $\gamma_0 = 10^{-6}$.

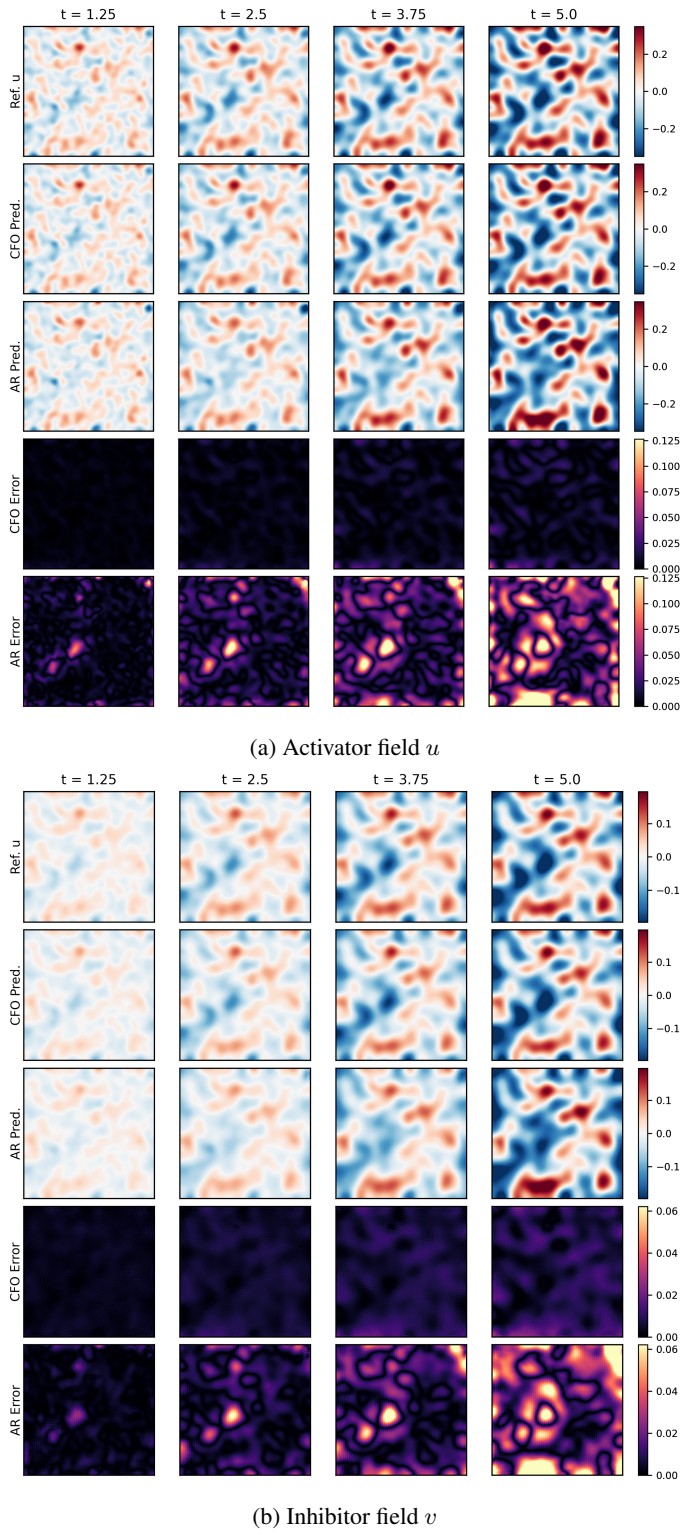

(a) Activator field $u$

(b) Inhibitor field $v$

Figure 5: DR trajectory visualization for activator (top) and inhibitor (bottom).

**Training of PDE-Refiner.** For the PDE-Refiner baseline, we follow the training setup from the original paper (Lippe et al., 2023) for 2D equations and use the 2D U-Net architecture with a parameter count and number of training steps matched to CFO for a fair comparison.

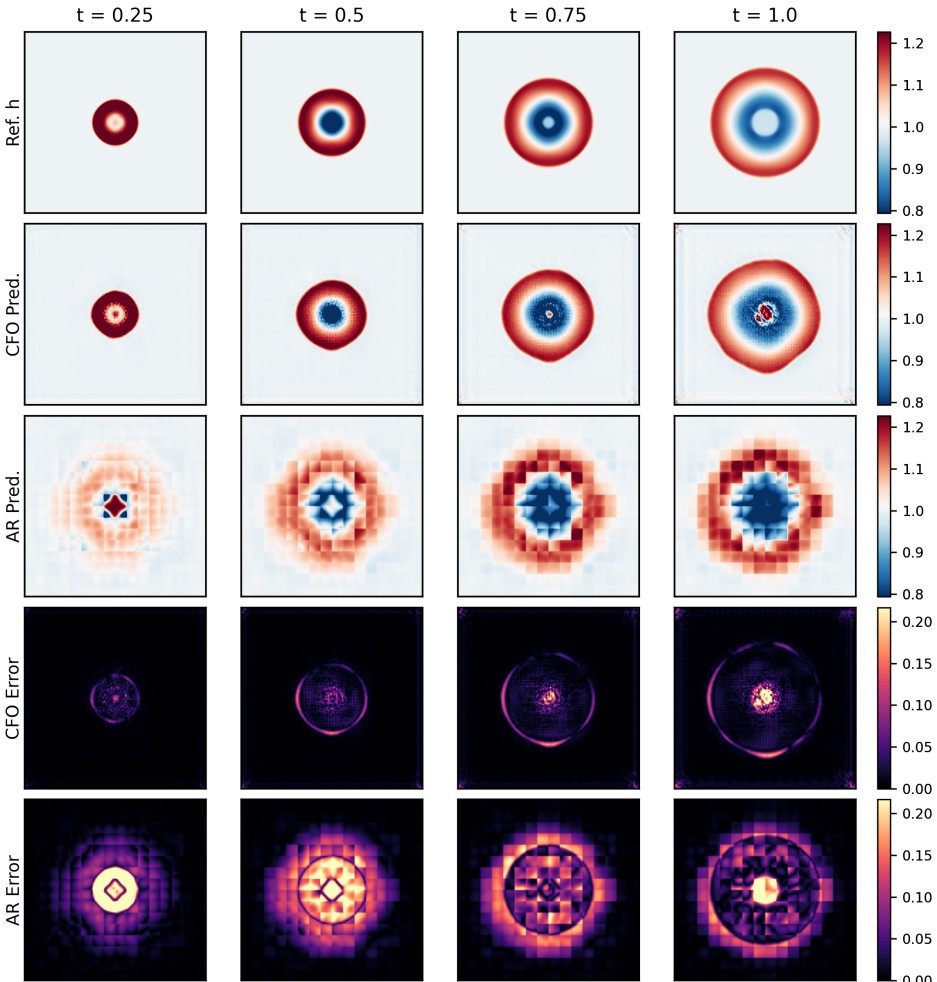

Figure 6: **Shallow Water Equation trajectory visualization.** Comparison of a long-horizon rollout for the test sample with the highest error. Quintic CFO, trained on 25% irregularly sampled data, captures the dynamics, while the autoregressive baseline (trained on full-resolution data) accumulates significant error.

### A.4.8 ABLATION EXPERIMENTS

To complement the ablation study in the main text, we provide additional results and figures in this subsection.

**Impact of noise schedule.** We sweep the noise magnitude $\gamma_0$. See Figure 7 (SWE on the left; Diffusion–Reaction on the right).

**Impact of ODE solvers and function evaluations.** We study the tradeoff between accuracy and computation by varying the solver and the number of function evaluations (NFE). Errors generally decrease with NFE, and higher-order solvers achieve lower error at a fixed budget. See the Euler and Heun sweeps in Tables 8 and 9 for final-time relative $L_2$ errors across tasks. Table 10 compares wall-clock inference time and accuracy between the AR baseline and CFO with RK4 solver using 50% of the AR NFE. Figure 8 visualizes the error–runtime across different NFE settings.

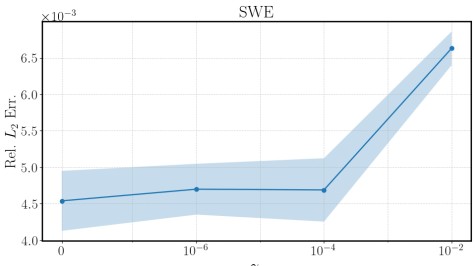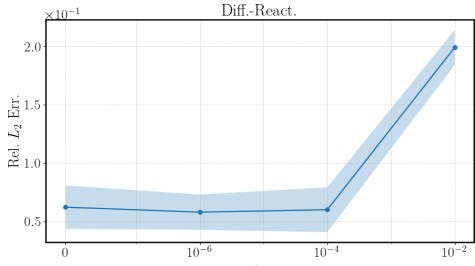

Figure 7: Impact of noise magnitude $\gamma_0$ on model performance for SWE (left) and DR (right) benchmarks.

Table 8: Influence of NFE on Quintic CFO (Euler solver) accuracy (Relative $L_2$ Error) at the final time step, compared to the autoregressive (AR) baseline.

| NFE (% AR) | Lorenz | Burgers | DR | SWE |
|---|---|---|---|---|
| AR (100%) | 0.1481 | 0.0647 | 0.3850 | 0.1048 |
| 50% | 0.3032 | 0.0142 | 0.0611 | 0.1345 |
| 100% | 0.2821 | 0.0111 | 0.0603 | 0.0978 |
| 200% | 0.2337 | 0.0099 | 0.0588 | 0.0651 |
| 400% | 0.2076 | 0.0094 | 0.0603 | 0.0389 |

Table 9: Influence of NFE on Quintic CFO (Heun solver) accuracy (Relative $L_2$ Error) at the final time step, compared to the autoregressive (AR) baseline.

| NFE (% AR) | Lorenz | Burgers | DR | SWE |
|---|---|---|---|---|
| AR (100%) | 0.1481 | 0.0647 | 0.3850 | 0.1048 |
| 50% | 0.1878 | 0.0090 | 0.0694 | 0.0874 |
| 100% | 0.1052 | 0.0090 | 0.0601 | 0.0249 |
| 200% | 0.0705 | 0.0090 | 0.0600 | 0.0076 |
| 400% | 0.0651 | 0.0090 | 0.0594 | 0.0061 |

**Comparison with other baselines.** Figure 9 shows the rescaled training loss for CFO and neural ODE for Lorenz system: CFO converges faster and more stably, especially in the early stage of training.

**Reverse-time inference.** We evaluate this the reverse-time inference capabilities of our framework on the dissipative Burgers' equation. Figure 10 visualizes the results.

**Temporal extrapolation beyond the training horizon.** We evaluate long-horizon temporal extrapolation by training on the first half of each trajectory, $t \in [0, T/2)$, and rolling out over the full horizon, $t \in [0, T]$. Table 11 reports the resulting errors, and Figure 11 shows error accumulation on the four benchmarks, compared to a baseline trained on the full time horizon.

**Neural Network Architectures.** CFO is largely architecture-agnostic: swapping the spatial backbone (FNO for 1D Burgers; DiT for 2D Diffusion–Reaction and SWE) yields comparable accuracy. See Table 12

**More Challenging Examples.** We further evaluate CFO on the one-dimensional Kuramoto–Sivashinsky (KS) equation, a fourth-order chaotic nonlinear PDE,

$$u_t + uu_x + u_{xx} + \nu u_{xxxx} = 0.$$

We follow the setting and dataset of (Majid, 2024) (140-step rollout) and use the same neural network architecture (1D U-Net) for all methods. As shown in Table 13, CFO achieves roughly $2\times$

| Task | Backbone | Time (ms/sample) | | Rel. $L_2$ Error | |
|---|---|---|---|---|---|
| | | AR | CFO (50% NFE) | AR | CFO (50% NFE) |
| Lorenz | MLP | 0.16 | **0.12** | $1.48 \times 10^{-1}$ | $9.19 \times 10^{-2}$ |
| Burgers | 1D U-Net | 0.68 | **0.31** | $1.03 \times 10^{1}$ | $9.00 \times 10^{-3}$ |
| DR | 2D U-Net | 117.18 | **49.36** | $3.74 \times 10^{-1}$ | $8.76 \times 10^{-2}$ |
| SWE | 2D U-Net | 111.83 | **49.63** | $3.84 \times 10^{-1}$ | $6.98 \times 10^{-2}$ |

Table 10: Inference runtime (ms/sample) and relative $L_2$ error at the final time step for the autoregressive baseline and CFO using 50% of the AR NFE, evaluated on a single NVIDIA A6000 GPU (JAX, fp32) with the same spatial backbones.

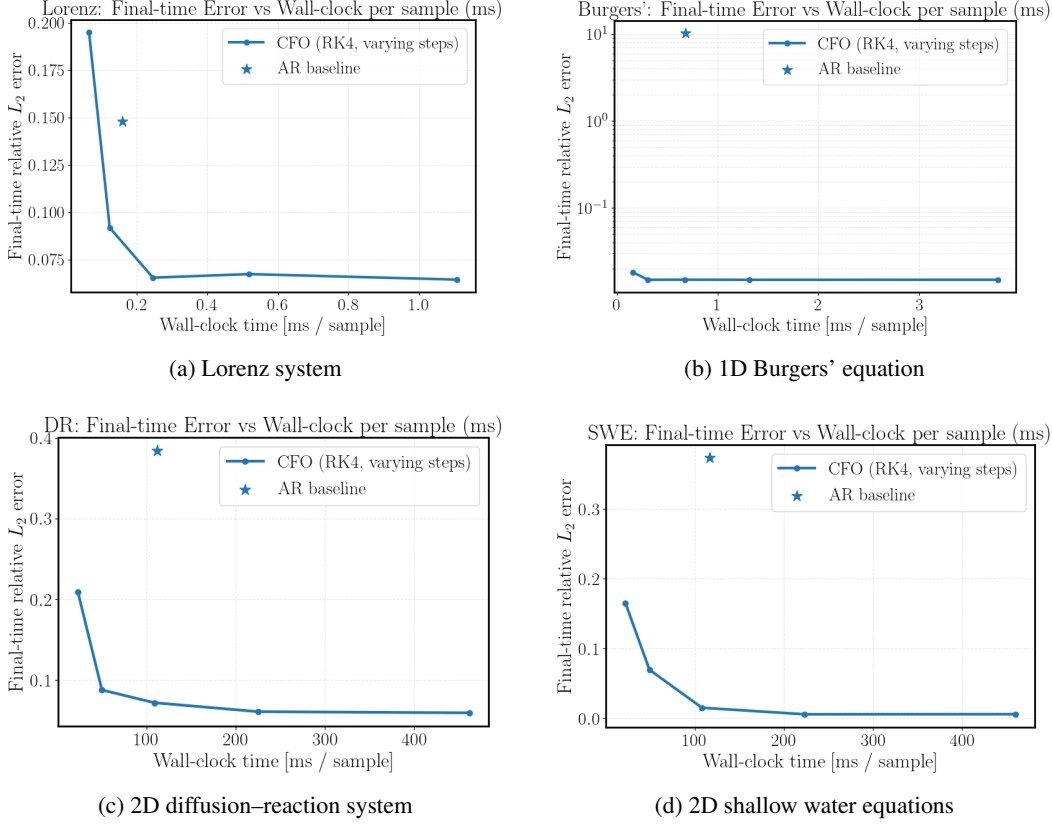

(a) Lorenz system

(b) 1D Burgers' equation

(c) 2D diffusion–reaction system

(d) 2D shallow water equations

Figure 8: Error versus wall-clock time for CFO with different numbers of function evaluations (NFE). The star marker denotes the AR baseline.

lower error than the neural ODE baseline and $3\times$ lower error than AR, while being vastly more computationally efficient than the neural ODE (comparable training time to AR).

**Latent Space Simulation.** In addition to direct application, CFO can be extended to operate in latent spaces. For the PDE in (2), consider an invertible operator $\mathcal{E}$ that maps the function $u$ to a latent representation $\mathcal{E}(u)$.

Define the conjugation of $\mathcal{N}$ as $\widetilde{\mathcal{N}} = \mathcal{E}\mathcal{N}\mathcal{E}^{-1}$, then the dynamics in the latent space can be expressed as:

$$\mathcal{E}(u_t) = \widetilde{\mathcal{N}}(\mathcal{E}(u)).$$

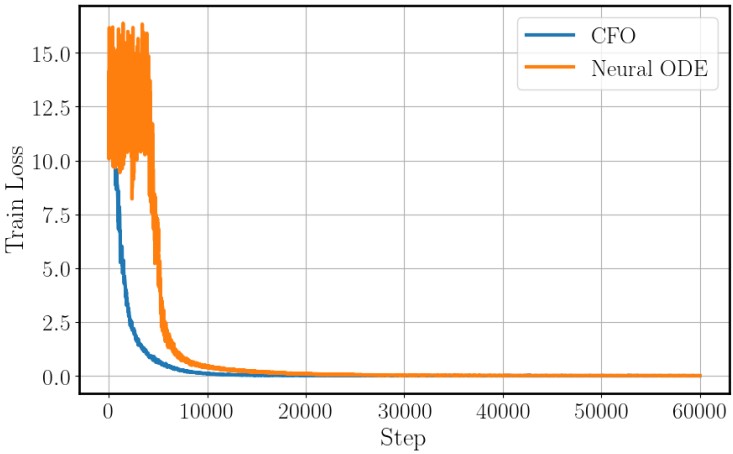

Figure 9: Training loss versus optimization steps for CFO and the neural ODE baseline on the Lorenz system. CFO converges faster and with substantially reduced oscillations, whereas the neural ODE exhibits large fluctuations in the early stages before eventually reaching a comparable loss level.

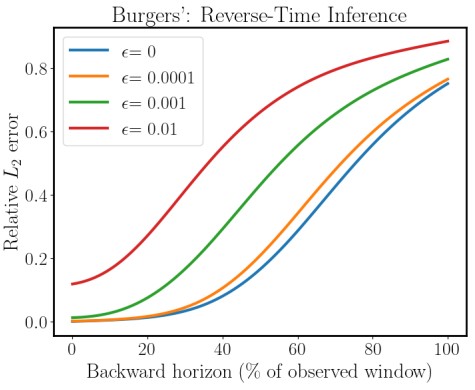

(a) Relative $L_2$ error of reconstructed $\hat{u}(t)$ for $t \in [0, 1)$ under different noise levels added to the terminal state $u(1)$.

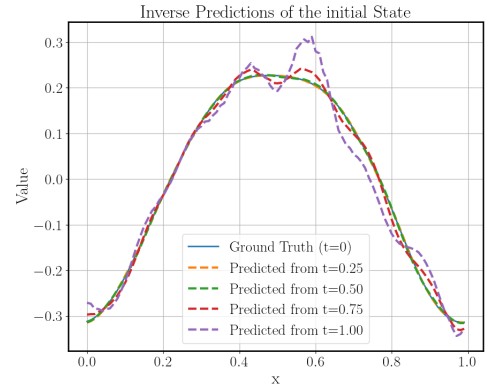

(b) Trajectories reconstructed backward from $t \in \{1, 0.75, 0.5, 0.25\}$ to $t = 0$.

Figure 10: Reverse-time inference for dissipative Burgers' equation.

Specifically, if $\mathcal{E}$ is a time-independent linear operator (e.g., Fourier or wavelet transform), then commuting with the time derivative yields:

$$[\mathcal{E}(u)]_t = \widetilde{\mathcal{N}}(\mathcal{E}(u)).$$

In this case, CFO learns the spatial operator $\widetilde{\mathcal{N}}$ directly in the latent space. Experiments on the 2D diffusion-reaction and 2D shallow water equations (see Table 14) show that CFO effectively models latent dynamics, achieving performance comparable to direct CFO in the original space.

| Equation | Train $[0, T]$ | Train $[0, T/2]$, Test $[0, T]$ |
|----------|----------------|----------------------------------|
| Lorenz | $4.53 \times 10^{-2}$ | $3.67 \times 10^{-2}$ |
| Burgers | $5.89 \times 10^{-3}$ | $7.94 \times 10^{-3}$ |
| DR | $4.37 \times 10^{-2}$ | $4.78 \times 10^{-2}$ |
| SWE | $4.56 \times 10^{-3}$ | $2.28 \times 10^{-2}$ |

Table 11: Relative $L_2$ error of temporal extrapolation beyond the training horizon.

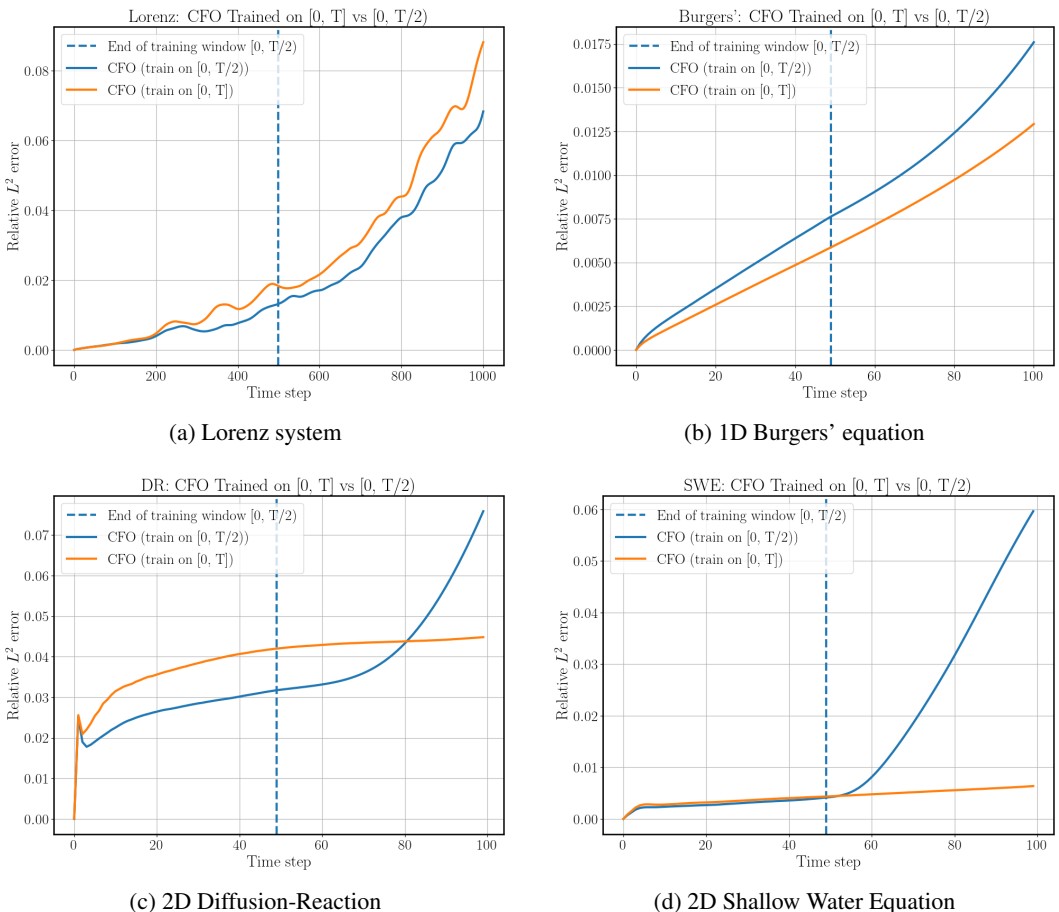

(a) Lorenz system                    (b) 1D Burgers' equation

(c) 2D Diffusion-Reaction            (d) 2D Shallow Water Equation

Figure 11: Long-horizon temporal extrapolation: error accumulation on Lorenz, Burgers, DR, and SWE. Models are trained on $t \in [0, T/2)$ and evaluated on rollouts over $t \in [0, T]$.

Table 12: CFO performance across (Relative $L_2$ Error) neural backbones—FNO (1D Burgers) and DiT (2D Diffusion–Reaction, 2D SWE).

| Equation | **Burgers** | **SWE** | **DR** |
|----------|-------------|---------|--------|
| Rel. $L_2$ Err. | 0.0068 | 0.0083 | 0.0821 |

| Method | Rel. $L_2$ Err. | Training Time |
|--------|-----------------|---------------|
| AR | 0.317 | 1 h |
| Neural ODE | 0.197 | 40 h 53 min |
| **CFO** | **0.105** | 1 h 5 min |

Table 13: Performance on the 1D Kuramoto–Sivashinsky (KS) equation. CFO attains substantially lower error than AR and Neural ODE while remaining far more computationally efficient than the Neural ODE baseline.

Table 14: CFO in wavelet latent space: Relative $L_2$ Error for 2D Shallow Water and 2D diffusion-reaction equations.

| Equation | SWE | DR |
|---|---|---|
| Rel. $L_2$ Err. | 0.0053 | 0.0218 |

