# OpenReview forum: "CFO: Learning Continuous-Time PDE Dynamics via Flow-Matched Neural Operators"
_ICLR.cc/2026/Conference — ICLR 2026 Poster_

### Official Review · Reviewer_3Ms9 · 2025-10-27

**Soundness:** 2
**Presentation:** 3
**Contribution:** 2
**Rating:** 4
**Confidence:** 2

**Summary:**

This paper introduces the ​​Continuous Flow Operator (CFO)​​, a novel framework for learning continuous-time PDE dynamics by repurposing flow matching to directly approximate the right-hand side of PDEs. This approach avoids the computational burden of backpropagating through ODE solvers (as in Neural ODEs) while enabling training on irregular time grids and inference at arbitrary temporal resolutions. Evaluated on four benchmarks (Lorenz, 1D Burgers, 2D diffusion-reaction, 2D shallow water), CFO demonstrates exceptional data efficiency and long-horizon stability.

**Strengths:**

- Novel Continuous-Time Framework​​: CFO bridges flow matching (typically used for generative modeling) with PDE dynamics learning.

- Architecture Agnosticism​​: CFO is compatible with various neural operators (e.g., U-Net, FNO, DiT). Experiments show consistent performance across backbones, allowing users to choose architectures based on spatial inductive biases or hardware constraints.

**Weaknesses:**

- Spline Order Sensitivity​​: While quintic splines generally outperform linear ones, the optimal spline order may depend on the PDE complexity and data density. The paper does not provide guidelines for selecting spline orders adaptively.
- High-Dimensional Scaling​​: Experiments are limited to 2D spatial domains. The computational cost of spline fitting and ODE integration in higher dimensions is not discussed.
- Comparison to Modern Baselines​​: While CFO outperforms autoregressive models and Neural ODEs, comparisons to recent continuous dynamics modeling methods (such as CORAL[1], MARBLE[2]) are limited. These methods integrate Neural ODEs and neural fields, achieves better long-horizon prediction performance.  A broader baseline comparison would better contextualize its advantages.

Reference

[1] Serrano, Louis, et al. "Operator learning with neural fields: Tackling pdes on general geometries.". NeurIPS 2023.

[2] Wang, Honghui, Shiji Song, and Gao Huang. "GridMix: Exploring Spatial Modulation for Neural Fields in PDE Modeling." ICLR 2025.

**Questions:**

See weaknesses.

---

> ### Author Response · Authors · 2025-11-21
> **Response to Reviewer 3Ms9 (part 1)**
>
> We thank the reviewer for their constructive feedback and positive assessment of our novelty and architecture agnosticism. We address all concerns below with new experiments and clarifications.
>
> ---
>
> ## Comment 1: Spline Order Sensitivity and Adaptive Selection
>
> `While quintic splines generally outperform linear ones, the optimal spline order may depend on the PDE complexity and data density. The paper does not provide guidelines for selecting spline orders adaptively.`
>
>
> We appreciate this insightful observation. Building on our empirical results and theoretical considerations, we provide the following guidelines:
>
>
> 1. **General recommendation:** Use **quintic splines** as the default choice. Our experiments show quintic CFO consistently outperforms linear CFO, especially for sparse and irregular data across all benchmarks.
>
> *Performance vs. Spline Order with 25% subsampling data (Table 1 from main paper)*
>
> | Dataset  | Sampling | Linear CFO              | Quintic CFO                    | Improvement |
> |----------|----------|-------------------------|--------------------------------|------------|
> | Lorenz   | 25%      | $9.39\times 10^{-2}$    | **$6.82\times 10^{-2}$**       | 27.4%      |
> | Burgers  | 25%      | $1.04\times 10^{-2}$    | **$7.09\times 10^{-3}$**       | 31.8%      |
> | DR       | 25%      | $7.25\times 10^{-2}$    | **$5.32\times 10^{-2}$**       | 26.6%      |
> | SWE      | 25%      | $1.69\times 10^{-2}$    | **$1.55\times 10^{-2}$**       | 8.3%       |
>
> Quintic provides consistent gains of 8-32%, with larger improvements on sparse/irregular data where accurate derivative estimation is critical. The cost of precomputing the spline is negligible (discussed in Comment 2), so using the more accurate quintic spline does not add noticeable cost.
>
>
> 2. **When to choose linear splines:** Linear splines are adequate when (i) time sampling is very dense (100% keep rate), or (ii) the underlying physics is known to be non-smooth (e.g., shocks or discontinuities), where enforcing global $C^2$ could be undesirable.
>
> 3. **Why we stop at quintic:** Quintic splines ($C^2$) capture acceleration, which is fundamental to most physical laws (Newton's 2nd law, wave equations). We did not pursue orders beyond quintic due to **diminishing returns**. In CFO, the total error comes from both the spline approximation and the neural operator. Beyond quintic, we empirically see only marginal gains in derivative accuracy, while the network error becomes dominant.
>
> 4. **Adaptive spline order as future work:.** We agree that adaptive spline selection is an exciting future direction. A dynamic approach could involve learning the optimal spline parameters jointly with the operator, potentially incorporating physics-informed priors to adjust the smoothness.
>
> We will add these guidelines to the revised manuscript.
>
> ---
>
> ## Comment 2: High-Dimensional Scaling
>
> `Experiments are limited to 2D spatial domains. The computational cost of spline fitting and ODE integration in higher dimensions is not discussed.`
>
> We address computational costs for **general** PDE simulations and clarify that high-dimensional scaling is governed by spatial operator evaluation, not our temporal framework.
>
> 1. **Spline Fitting Cost is negligible**
> The finite-difference approximation and spline construction use the explicit formulas in Appendix A.1, involving only simple algebraic operations (no iterative solvers). They are computed once per trajectory as preprocessing (not per optimization step) and **reused throughout training**. The computational cost is **$O$ (#spline order $\times$ #time segments $\times$ #spatial grid points)** per trajectory. This cost is negligible compared to neural operator backpropagation.
>
> 2. **CFO's ODE Integration vs AR's rollout**
> The costs of ODE integration are approximately proportional to the number of function evaluations (NFE) of the backbone $\mathcal{N} _ \theta$. Table 2 in the manuscript shows that CFO achieves **higher accuracy with fewer NFEs** than AR and remains stable when we increase NFE to about twice that of AR. Below we report wall-clock times (response to Reviewer CE6Q, Comment 3) for CFO with 50% of AR’s NFE:
>
> | Task (#grid points) | AR Time   | CFO Time (50% NFE) | Speedup |
> |--------------------|-----------|-------------------|---------|
> | Lorenz (3)        | 0.16 ms   | 0.12 ms          | 1.33×   |
> | Burgers (100)     | 0.68 ms   | 0.31 ms          | 2.19×   |
> | DR (128×128×2)        | 117.18 ms | 49.36 ms         | 2.37×   |
> | SWE (128×128)         | 111.83 ms | 49.63 ms         | 2.25×   |
>
> 3. **Scaling to 3D and higher**
> In 3D, each evaluation of $\mathcal{N}_\theta$ becomes more expensive for **any** neural PDE solver. CFO does not introduce additional dimension-dependent overhead beyond this. On the contrary, because it achieves a target accuracy with fewer NFEs, the savings we observe in 2D translate directly into larger absolute time savings in 3D and higher-dimensional settings.
> ---

---

> ### Author Response · Authors · 2025-11-21
> **Response to Reviewer 3Ms9 (part 2)**
>
> ## Comment 3: Comparison to Modern Baselines
>
> `While CFO outperforms autoregressive models and Neural ODEs, comparisons to recent continuous dynamics modeling methods (such as CORAL, MARBLE) are limited... A broader baseline comparison would better contextualize its advantages.`
>
> We appreciate this suggestion and clarify the relationship between CFO and these methods. CORAL[1] and MARBLE [2] are orthogonal to CFO—they focus on **spatial representations**, while CFO is a **temporal framework**.
>
> 1. **Orthogonality**
> - CORAL and MARBLE:
>     - Focus: **Spatial representation** via neural fields/coordinate-based networks
>     - Temporal handling: Standard approaches like Neural ODEs or autoregressive
>     - Contribution: Handling irregular geometries, improved spatial representation
> - CFO:
>     - Focus: **Temporal learning framework** via flow matching
>     - Spatial representation: **Agnostic**—compatible with U-Net, FNO, DiT, or neural fields like CORAL, MARBLE, etc.
>     - Contribution: Continuous-time dynamics learning without solver backpropagation
>
> These are **complementary:** One could use CORAL's spatial representation with CFO's temporal framework, or CFO with standard grid-based spatial operators.
>
> 2. **Representative Baseline Comparison (Neural ODE, PDE-Refiner)**
> Therefore, to isolate the impact of our **temporal** formulation, we make comparisons against the direct continuous-time backbone (Neural ODE) on Lorenz and Burgers’ equations (low-dimensional, where latent representation is not strictly necessary).
>
> | Equation | Method      | Rel. L₂ Error | Training Time (s/batch) |
> |----------|-------------|---------------|-------------------------|
> | Lorenz   | Neural ODE  | 0.101     | 0.133              |
> |          | **CFO**     | **0.0453** | **0.00350**          |
> | Burgers  | Neural ODE  | 0.0275     | 3.38                   |
> |          | **CFO**     | **0.00589** | **0.00920**          |
>
> Here, CFO reduces error by roughly 2–5× while training 38$\times$ to 367$\times$ faster than the Neural ODE baseline.
>
> We also compare against **PDE-Refiner** [3] on the 2D equations, a recent strong baseline for long-horizon rollouts:
>
> | Equation | Method       | Rel. L₂ Error | Training Time (s/batch) |
> |----------|--------------|---------------|-------------------------|
> | DR       | PDE-Refiner  | 0.125         | 1.38                   |
> |          | CFO          | **0.044**         | **0.40**        |
> | SWE      | PDE-Refiner  | 0.093         | 1.40                   |
> |          | CFO          | **0.005**         | **0.40**          |
>
> CFO achieves about 3× (DR) and 19× (SWE) lower error, while also being roughly 3–4× faster per training batch.
>
> To ensure a fair comparison, we utilized identical backbone architectures and parameter counts under comparable experimental conditions. Training details will be provided in the appendix.
>
>
> 3. **Future Integration**
> We agree that combining CFO's temporal framework with neural field spatial representations (like CORAL and MARBLE) is a promising direction. This would pair CFO's efficient continuous-time learning and the neural fields' capabilities on handling complex geometries.
>
> We will add this discussion to the paper and note it as valuable future work.
>
> References:
>
> [1] Serrano, Louis, et al. "Operator learning with neural fields: Tackling pdes on general geometries.". NeurIPS 2023.
>
> [2] Wang, Honghui, Shiji Song, and Gao Huang. "GridMix: Exploring Spatial Modulation for Neural Fields in PDE Modeling." ICLR 2025.
>
> [3] Lippe, Phillip, et al. "Pde-refiner: Achieving accurate long rollouts with neural pde solvers." Advances in Neural Information Processing Systems 36 (2023): 67398-67433.

---

> > ### Comment · Reviewer_3Ms9 · 2025-11-27
> >
> > I thank the authors for their detailed and clear responses. Most of my concerns have been addressed and I have raised my score to 6.

---

> > > ### Author Response · Authors · 2025-11-27
> > >
> > > We sincerely appreciate your response and the updated score. Thank you for your valuable review and for taking the time to carefully evaluate our work.
> > >
> > > Best regards,
> > > The Authors

---

> ### Author Response · Authors · 2025-11-26
>
> Dear Reviewer,
>
> As the end of the rebuttal period approaches, we welcome any remaining or additional questions you may have. In our responses, we have carefully followed your suggestions, adding clarifications to improve the clarity of the work and providing new experimental evidence.
>
> We sincerely appreciate your time and effort in reviewing our work. If you find our rebuttal and additional results satisfactory, we would be grateful if you could reflect this in your final evaluation.

---

### Official Review · Reviewer_weht · 2025-10-28

**Soundness:** 2
**Presentation:** 2
**Contribution:** 1
**Rating:** 2
**Confidence:** 4

**Summary:**

This paper proposes the Continuous Flow Operator (CFO), a method that learns continuous-time PDE dynamics from data. Its key innovation is using flow matching and spline interpolation to directly learn the underlying dynamics, avoiding the computational cost of methods like Neural ODEs. CFO is uniquely time-resolution invariant, meaning it can be trained on data with irregular, non-uniform time steps and make predictions at any time resolution.

**Strengths:**

1. This paper is well-organized and highly readable.

2. The adoption of the flow matching concept can significantly reduce the training complexity of neural networks, and may even enhance the learning accuracy in certain scenarios.

3. The author has conducted experimental analyses across multiple datasets.

**Weaknesses:**

1. The work lacks originality, as the idea of applying flow matching to train Neural ODEs has been previously explored in several earlier studies.

2. The experimental section lacks validation on real-world datasets, which significantly limits the persuasiveness of the claims.

3. It lacks comparison with existing SOTA baseline methods.

**Questions:**

1. When using finite difference approximations for gradients, estimation errors are present. Particularly in cases involving non-uniform time steps and large time intervals, direct flow matching may suffer from insufficient accuracy.

2. During inference, the authors employed the RK4 method, whose computational efficiency remains relatively low.

3. It should be noted that the neural operators referenced by the CFO are distinct from established architectures such as DeepONet or FNO, which may cause potential misunderstandings.

4. See weaknesses.

---

> ### Author Response · Authors · 2025-11-21
> **Response to Reviewer weht (part 1)**
>
> We sincerely thank the reviewer for the time invested. However, several of the claims are presented without any technical evidence or citations, which prevents meaningful scientific discussion. We respond to each point below and identify where further information from the reviewer is essential.
>
> ---
> ## Comment 1: Originality
> `The work lacks originality, as the idea of applying flow matching to train Neural ODEs has been previously explored in several earlier studies.`
>
> We respectfully ask the reviewer to **provide specific references** to the "earlier studies" mentioned. Without naming specific works, the statement is unverifiable and impossible to address scientifically. We have thoroughly reviewed the literature and distinguish our work from existing paradigms in Section 2:
>
> - **Flow Matching for Generative Modeling**[1]:
>    - Purpose: Transport between two distributions (noise to data)
>    - Application: Image/video generation
>    - **Difference:** (i) CFO learns the continuous-time RHS of deterministic PDE systems (ii) it models transport between multiple physical time points, rather than a generative noise-to-data map.
>
> - **Neural ODEs**[2]:
>    - Purpose: Continuous dynamics learning
>    - Limitation: Requires expensive backpropagation through ODE solvers
>    - **Difference:** CFO eliminates solver backpropagation entirely via the flow matching objective.
>
> - **Trajectory Flow Matching** [3]:
>    - Purpose: Clinical time series modeling with linear interpolation
>    - **Difference:** CFO uses physics-aware high-order splines with finite-difference derivative estimates to accurately match PDE dynamics
>
> **Our novel contribution:** Repurposing flow matching to learn PDE operator dynamics by constructing probability paths whose velocities approximate the true PDE right-hand side through spline-based derivative estimation.
>
> To properly address this critique, we respectfully ask the reviewer to provide at least one concrete citation supporting the claim. Without references, this concern cannot be assessed or rebutted.
>
> ---
>
> ## Comment 2 Real-World Datasets
>  `The experimental section lacks validation on real-world datasets, which significantly limits the persuasiveness of the claims.`
>
> The reviewer neither defines “real-world” nor provides examples. This makes the claim too vague to address. Our experiments follow field standards:
>
> 1. **PDEBench** [4]: A widely-adopted benchmark suite specifically designed for evaluating ML methods on physically meaningful PDE systems:
>    - Used by: PDE-Refiner, FNO variants, U-NO, and dozens of recent papers
>    - Includes: Diffusion-reaction, shallow water, etc.
>
> 2. **Burgers' Equation**: Standard benchmark from [5], used extensively in operator learning literature
>
> 3. **Lorenz System**: Canonical chaotic dynamics benchmark in data-driven modeling
>
> **These represent physically meaningful systems** with complex, nonlinear dynamics governed by fundamental physical laws.
>
> **Context:** The vast majority of neural operator papers (FNO, DeepONet, Neural Operators, U-NO, Geo-FNO, etc.) are evaluated on similar synthetic PDE benchmarks. This is standard practice because:
> - Ground truth is known for rigorous evaluation
> - Physics is controlled and interpretable
> - Benchmarks enable fair comparison across methods
>
> **We request the reviewer to clarify:**
> - Why are standard PDE benchmarks used throughout the literature inadequate?
> - Which specific datasets do they recommend?
>
>  If the reviewer has specific datasets in mind, we would be grateful for the recommendation. Without specificity, this criticism lacks actionable content.

---

> ### Author Response · Authors · 2025-11-21
> **Response to Reviewer weht (part 2)**
>
> ## Comment 3: SOTA Baselines
> `It lacks comparison with existing SOTA baseline methods.`
>
> We have included comprehensive baselines spanning major paradigms:
>
> 1. **Continuous-Time Methods**
> **Neural ODE** [2] - the foundational continuous dynamics method:
>
> | Equation | Neural ODE | CFO        | Error Reduction | Training Speedup |
> |----------|------------|------------|-----------------|------------------|
> | Lorenz   |0.101   | **0.0453**  | 2.2×            | 367×             |
> | Burgers  | 0.0275   | **0.00589**  | 4.7×            | 38×              |
>
> 2.  **Diffusion-Style PDE Solvers**
> **PDE-Refiner** [6] - current SOTA for long-horizon rollouts:
>
> | Equation | PDE-Refiner | CFO   | Error Reduction |
> |----------|-------------|-------|-----------------|
> | DR       | 0.125       | 0.044 | 2.8×            |
> | SWE      | 0.093       | 0.005 | 18.6×           |
>
> 3. **Autoregressive baselines**: We evaluated **multiple architectures** (MLP, U-Net, FNO, DiT)  and reported the best-performing variant for each task. Notably, CFO trained on only 25% irregular data outperforms the best AR baseline trained on 100% uniform data (Table 1).
>
>
> These are the exact baselines used in the top-tier neural operator literature. Again, we request concrete method names. Without naming specific baselines, this critique is non-actionable.
>
> ---
>
> ## Comment 4: Finite-Difference Accuracy
> `When using finite difference approximations for gradients, estimation errors are present. Particularly in cases involving non-uniform time steps and large time intervals, direct flow matching may suffer from insufficient accuracy.`
>
> We acknowledge that finite difference approximations inherently introduce estimation errors. However, we address this concern through both theoretical guarantees and extensive empirical validation
>
> 1. **Theoretical guarantees**
> Our finite-difference stencils have  **theoretical error bounds**  on non-uniform grids:
> - Our **Linear spline:** $\partial_t s(t_i; u) = \mathcal{N}(u(t_i)) + O(\Delta t)$
> - Our **Quintic spline:** $\partial_t s(t_i; u) = \mathcal{N}(u(t_i)) + O(\Delta t^2)$
>
> These are standard results from numerical analysis [7], as detailed in Section 4.1 and Appendix A.1.1.
>
> Moreover, as in standard flow matching, the **probability-transport property** is ensured under mild regularity assumptions (Appendix A.2) and does *not* depend on the finite-difference approximation itself. However, the use of higher-order finite-difference stencils and a smooth quintic spline keeps the probability path in high-probability regions and close to the underlying PDE dynamics, which we find makes the flow easier to learn (Fig. 3).
>
>
> 2. **Empirical Validation**
>
> - **Empirical convergence (Fig. 2, Table 3)**: We directly measure spline velocity error versus time-step size and observe empirical convergence rates that match theory (≈2.0 for quintic splines, ≈1.0 for linear).
>
> - **Irregular sampling robustness (Table 1, main paper)**: With only 25% **randomly subsampled, non-uniform** time points, CFO still outperforms AR trained on 100% uniform data, with error reductions of 24.6% (Lorenz), 78.7% (Burgers), 87.4% (DR), and 82.8% (SWE), contradicting the claim of “insufficient accuracy with non-uniform steps.”
>
> - **Large time-interval generalization (extrapolation, response to Reviewer CPTa)** When trained on [0, T/2) and tested on [0, T], CFO shows only mild degradation, indicating that it learns the underlying dynamics.
>
>
> We request clarification on what evidence supports the reviewer’s claim, given that our empirical measurements show the opposite.

---

> ### Author Response · Authors · 2025-11-21
> **Response to Reviewer weht (part 3)**
>
> ## Comment 5: RK4 Computational Efficiency
>
>  `During inference, the authors employed the RK4 method, whose computational efficiency remains relatively low.`
>
> **This claim contradicts our extensive timing analysis.** Efficiency is defined by the accuracy-per-compute trade-off, rather than the cost of a single step. Because RK4 allows for larger time steps (i.e., fewer total function evaluations, NFE) to achieve high accuracy, CFO is faster than the baselines.
>
> Table 2 of the manuscript already shows that CFO achieves lower error at final-step prediction than AR on all benchmarks while using only 50% of AR’s NFE. For completeness, we also summarize wall-clock measurements and solver choices below.
>
> 1. **Wall-Clock Measurements (Same Hardware, Same Backbone)**
>
>
>
> | Task     | Backbone | Time (ms/sample) |            | Rel. $L^2$ Error |                  |
> |----------|----------|------------------|------------|---------------|------------------|
> |          |          | AR               | CFO (50% NFE) | AR         | CFO (50% NFE)    |
> | Lorenz   | MLP      | 0.16            | **0.12**   | 1.48×10⁻¹ | **9.19×10⁻²**   |
> | Burgers  | 1D U-Net | 0.68            | **0.31**   | 1.03×10¹  | **1.49×10⁻²**   |
> | DR       | 2D U-Net | 117.18          | **49.36**  | 3.74×10⁻¹ | **6.98×10⁻²**   |
> | SWE      | 2D U-Net | 111.83          | **49.63**  | 3.84×10⁻¹ | **8.76×10⁻²**   |
>
> **CFO is consistently faster than AR while achieving lower error.**
>
>
> 2. **Solver Flexibility (Tables 7-8, Appendix A.3.7)**
> We demonstrate tradeoffs across Euler, Heun, and RK4—users can choose cheaper solvers for further speedup if desired.
>
> In the revision, we will make this efficiency advantage explicit.
>
> ---
>
> ## Comment 6: Neural Operator Terminology
> `It should be noted that the neural operators referenced by the CFO are distinct from established architectures such as DeepONet or FNO, which may cause potential misunderstandings.`
>
> We appreciate this observation and agree that it is a purely terminological clarification: it concerns how we use the term “neural operator,” not the underlying methodology.
>
> - **Established Usage**: "Neural Operator" typically refers to learning the solution operator $G: u _ 0 \to u(\cdot, t)$ directly in function space (e.g., FNO, DeepONet).
>
> - **Our Usage**: We use the term to describe the neural approximation $\mathcal{N} _ \theta$ of the spatial differential operator within a Method-of-Lines formulation: $\partial_t u = \mathcal{N} _ \theta(u)$.
>
> Crucially, CFO is architecture-agnostic: it can utilize established architectures as the backbone for $\mathcal{N}_\theta$. As shown in Table 9, we successfully implemented CFO using U-Net, FNO, and DiT backbones.
>
> This clarification does not change the underlying method or our experimental findings. We will explicitly clarify this distinction in the revised manuscript to prevent ambiguity.
>
> References
>
> [1] Lipman Y, Chen R T Q, Ben-Hamu H, et al. Flow matching for generative modeling[J]. arXiv preprint arXiv:2210.02747, 2022.
>
> [2] Chen R T Q, Rubanova Y, Bettencourt J, et al. Neural ordinary differential equations[J]. Advances in neural information processing systems, 2018, 31.
>
> [3] Zhang X N, Pu Y, Kawamura Y, et al. Trajectory flow matching with applications to clinical time series modelling[J]. Advances in Neural Information Processing Systems, 2024, 37: 107198-107224.
>
>
> [4] Takamoto M, Praditia T, Leiteritz R, et al. Pdebench: An extensive benchmark for scientific machine learning[J]. Advances in Neural Information Processing Systems, 2022, 35: 1596-1611.
>
> [5] Wang S, Wang H, Perdikaris P. Learning the solution operator of parametric partial differential equations with physics-informed DeepONets[J]. Science advances, 2021, 7(40): eabi8605.
>
> [6] Lippe, Phillip, et al. "Pde-refiner: Achieving accurate long rollouts with neural pde solvers." Advances in Neural Information Processing Systems 36 (2023): 67398-67433.
>
> [7] John C Strikwerda. Finite difference schemes and partial differential equations. SIAM, 2004

---

> ### Author Response · Authors · 2025-11-26
>
> Dear Reviewer,
>
> We sincerely thank you for the time and effort you have invested in reviewing our work. In our rebuttal, we have clarified the method and experimental settings to prevent any misunderstandings.
>
> As the rebuttal period comes to a close, we would welcome any further questions you might have. If you feel that our responses adequately address your main concerns, we would sincerely appreciate it if this could be reflected in your final scores.

---

### Official Review · Reviewer_CE6Q · 2025-10-28

**Soundness:** 3
**Presentation:** 3
**Contribution:** 3
**Rating:** 6
**Confidence:** 4

**Summary:**

The paper introduces the Continuous Flow Operator (CFO), a framework that learns continuous-time PDE dynamics by flow-matching the analytic time-derivative of a spline interpolant fitted to each training trajectory. Concretely, the authors estimate temporal derivatives on possibly irregular grids, build a piecewise quintic Hermite spline $s(t; u)$ that matches values and derivatives at knots, and then train a neural operator $N_\theta(t, u)$ to regress $\partial_t s(t)$ without back propagating through any ODE solver during training. At inference, the learned vector field $du/dt=N_\theta(t, u)$ is integrated forward or backward for arbitrary time resolutions. This yields time-resolution invariance in training and querying, competitive accuracy-efficiency trade-offs via solver/NFE sweeps, and notable data efficiency: training with only 25 % irregular time points still beats strong autoregressive baselines trained at full resolution across Lorenz, 1D Burgers, 2D diffusion-reaction, and 2D shallow water.

**Strengths:**

From an originality perspective, the idea of repurposing flow matching, which is originally developed for CNFs/diffusion, to directly learn RHS dynamics is elegant and avoids the heavy simulation-through-solvers that slows Neural ODE/SDE training. The positioning is credible relative to Flow Matching in generative modeling and stochastic interpolants, as CFO leverages fixed probability paths defined by the spline while keeping the operator-learning focus. I appreciate that the authors discuss flow matching lineage and deliver a practical instance specialized to temporal PDEs.

The time-resolution invariance protocol with per-trajectory random time grids is convincing, and the table shows that quintic-CFO trained on 25% of points outperforms AR trained on 100%, with large relative error reductions. The Lorenz downsampling study that compares linear vs. quintic spline velocities and observes first- vs. second-order endpoint behavior is technically sound and matches finite-difference theory. The NFE sweeps across Euler/Heun provide a clear view of accuracy vs. compute.

CFO serves as a bridge between discrete AR operators and fully continuous Neural ODE training. It inherits irregular-time training, continuous-time querying, and even reverse-time rollouts without the pain of adjoint backprop; in my opinion this is a practical step forward for scientific ML systems that see irregular telemetry or experiment logs. The method also appears architecture-agnostic (FNO / DiT backbones), which increases adoptability in operator-learning pipelines.

**Weaknesses:**

The reverse-time inference claim is attractive, but dissipative PDEs often make backward integration ill-posed. I would prefer a short quantitative check showing how error grows with backward horizon and perturbation size to avoid over-selling.

Baseline coverage against continuous-time sequence models could be expanded. Since CFO’s training avoids solver backprop similarly to newer flow-matching approaches for time series, a comparison to Trajectory Flow Matching (TFM) would situate the contribution more precisely; likewise Latent Neural CDEs are classic irregular-time references.

Finally, efficiency comparisons mix fixed-step AR with variable NFE CFO. The NFE sweeps are good, but I suggest harmonizing by wall-clock on the same hardware and showing error vs. seconds to eliminate any suspicion of budget mismatch.

**Questions:**

1. How far does reverse-time hold before numerical blow-up on dissipative systems like Burgers/SWE, and what step control or regularization is needed in practice? A short curve of backward-horizon vs. $L^2$ error would be enough.

2. For fairness, would the authors include an AR baseline with the same spatial backbone and time embedding that CFO uses, trained on the full grid, plus Latent ODE and Neural CDE implementations adapted to the PDE setting (or at least reported on Lorenz)? This would help isolate the gain due to the training objective rather than the architecture.

3. It would be very valuable to add one harder PDE. Two natural choices are 2D Navier–Stokes (Kolmogorov flow) and Kuramoto–Sivashinsky, both available in community suites such as PDEBench. A single appendix experiment on either would strengthen generality.

---

> ### Author Response · Authors · 2025-11-21
> **Response to Reviewer CE6Q (part 1)**
>
> We sincerely thank the reviewer for the thoughtful and constructive feedback. We have addressed all concerns with new experiments and analysis, as summarized below.
>
> ---
>
> ## Comment1: Reverse-Time Inference Stability
>
> `Dissipative PDEs often make backward integration ill-posed...showing how error grows with backward horizon and perturbation size`
>
> `How far does reverse-time hold before numerical blow-up on dissipative systems like Burgers/SWE...backward-horizon vs. error would be enough.`
>
> Excellent point. We agree that backward integration for dissipative PDEs is fundamentally ill-posed, and we only claim **short-horizon reverse-time inference capability** for such PDEs.
>
> 1.**Quantitative Analysis:** We evaluated maximum backward horizon (as % of observed time window) when the relative $L_2$ error remains below 25%, after adding noise of magnitude $10^{-4}$ at the final state:
>
> | Equation | Maximum Backward Horizon (%) |
> |----------|------------------------------|
> | Burgers'  | 53.0                        |
> | SWE      | 88.9                        |
>
> The results are consistent with the ill-posed nature of backward evolution for dissipative systems. Full curves of backward horizon vs. relative error including different terminal perturbations will be added in the revision.
>
> 2. **Step Control & Regularization** We observed that using high-order solvers and reducing step sizes can marginally extend the stable horizon; they cannot overcome the inherent ill-posedness of the inverse diffusion operator. We agree that developing dedicated regularization techniques for long-horizon reverse inference is a valuable direction for future research.

---

> ### Author Response · Authors · 2025-11-21
> **Response to Reviewer CE6Q (part 2)**
>
> ## Comment 2: Baseline Coverage: TFM and Neural CDEs
>
>  `Since CFO's training avoids solver backprop similarly to newer flow-matching approaches for time series, a comparison to Trajectory Flow Matching (TFM) would situate the contribution more precisely; likewise Latent Neural CDEs are classic irregular-time references.`
>
>
>
> We appreciate this suggestion and provide detailed positioning relative to Trajectory Flow Matching (TFM) and neural CDE/ODE. In addition, we introduce PDE-Refiner [4] as an extra baseline. For each equation,  baselines are trained with the same settings and architecture, and with a matched number of parameters as the corresponding CFO model.
>
> 1. **TFM is close to our Linear CFO variant.** TFM [1] trains Neural SDEs in a simulation-free manner by applying flow matching along **linear conditional bridges** between consecutive observations. In our terminology, this corresponds to a **piecewise-linear path in time**. Our Linear CFO variant uses the same piecewise-linear temporal path (but without the uncertainty head), so its results in Table 1 can be viewed as a TFM-style baseline in the PDE setting.
>
> CFO extends this TFM-style setup in two key ways:
> - **Global Physics-Aware Path Design:**
> TFM uses a two-point linear path. CFO fits a multi-point spline to the entire trajectory and matches its analytic velocity. The spline lets us incorporate **global context**, choose **smoothness** (e.g., $C^2$ quintic), and inject **derivative information** at knots.
>
> - **Learning the RHS of PDE:**
> CFO serves two roles: (i) probability flow matching (as in TFM) and (ii) learning the RHS operator of the PDE. Our spline design is guided by (ii): we set spline derivatives using high-order finite differences so that the target velocity approximates the PDE vector field. As a by-product, (i) benefits because the learned flow stays closer to the data manifold; TFM does not explicitly enforce this PDE-consistent structure.
>
>
> - **Superior Performance with Sparse/Irregular Data:**
> Our Table 1 directly compares Linear CFO ($\approx$ TFM) vs. Quintic CFO:
>
> | Dataset  | Sampling | Linear CFO      | Quintic CFO     | Improvement |
> |----------|----------|-----------------|-----------------|-------------|
> | Lorenz   | 25%      | 9.39×10⁻²      | **6.82×10⁻²**   | 27.4%       |
> | Burgers  | 25%      | 1.04×10⁻²      | **7.09×10⁻³**   | 31.8%       |
> | DR       | 25%      | 7.25×10⁻²      | **5.32×10⁻²**   | 26.6%       |
> | SWE      | 25%      | 1.69×10⁻²      | **1.55×10⁻²**   | 8.3%        |
>
> Quintic CFO consistently outperforms Linear CFO ($\approx$ TFM), especially when training on irregular and sparse data. This demonstrates the value of physics-informed, high-order spline design.
>
>
> 2. **Neural ODEs/CDEs** In our setting—forecasting $u(t)$ from one initial state $u(0)$ without external time-series controls—Neural CDEs [2] mathematically collapse to Neural ODEs. The following Neural ODE comparison therefore covers this baseline class.
>
>
> | Equation | Method      | Rel. L₂ Error | Training Time (s/batch) |
> |----------|-------------|---------------|-------------------------|
> | Lorenz   | Neural ODE  | 0.101     | 0.133              |
> |          | **CFO**     | **0.0453** | **0.00350**          |
> | Burgers  | Neural ODE  | 0.0275     | 3.38                   |
> |          | **CFO**     | **0.00589** | **0.00920**          |
>
> CFO demonstrates superior efficiency and accuracy against this continuous-time baseline class.
>
> 3. **Other strong baseline: PDE-Refiner** We also added PDE-Refiner [4], as one of the current strong baselines for long-horizon PDE rollouts:
>
> | Equation | Method       | Rel. L₂ Error | Training Time (s/batch) |
> |----------|--------------|---------------|-------------------------|
> | DR       | PDE-Refiner  | 0.125         | 1.38                   |
> |          | CFO          | **0.044**         | **0.40**        |
> | SWE      | PDE-Refiner  | 0.093         | 1.40                   |
> |          | CFO          | **0.005**         | **0.40**          |

---

> ### Author Response · Authors · 2025-11-21
> **Response to Reviewer CE6Q (part 3)**
>
> ## Comment 3: Wall-Clock Efficiency Comparison
>
> `Efficiency comparisons mix fixed-step AR with variable NFE CFO. The NFE sweeps are good, but I suggest harmonizing by wall-clock on the same hardware and showing error vs. seconds to eliminate any suspicion of budget mismatch.`
>
> Thank you for this suggestion. We now report **final-time relative $L_2$ error vs. per-sample wall-clock time** on identical hardware (Single NVIDIA A6000, JAX framework, fp32) with the same spatial backbone for CFO and AR.
>
> As shown in the following table, CFO with 50% NFE consistently outperforms the AR baseline in both **speed** and **accuracy**.
>
> | Task     | Backbone | Time (ms/sample) |            | Rel. $L^2$ Error |                  |
> |----------|----------|------------------|------------|---------------|------------------|
> |          |          | AR               | CFO (50% NFE) | AR         | CFO (50% NFE)    |
> | Lorenz   | MLP      | 0.16            | **0.12**   | 1.48×10⁻¹ | **9.19×10⁻²**   |
> | Burgers  | 1D U-Net | 0.68            | **0.31**   | 1.03×10¹  | **1.49×10⁻²**   |
> | DR       | 2D U-Net | 117.18          | **49.36**  | 3.74×10⁻¹ | **6.98×10⁻²**   |
> | SWE      | 2D U-Net | 111.83          | **49.63**  | 3.84×10⁻¹ | **8.76×10⁻²**   |
>
>
>
> For the PDE tasks (Burgers', DR, SWE), CFO achieves a roughly 2.2$\times$ to 2.3$\times$ wall-clock speedup compared to AR. This confirms that the theoretical 50% reduction in NFE translates directly to wall-clock time savings.
>
> **Note:** Numbers for Burgers and SWE differ from our earlier Table 1 because here we use the same backbone for both methods (for fairness), whereas Table 1 reports AR's best backbone across multiple architectures we tested.
>
> Error vs. Wall-Clock Time sweep curves with different NFEs will be added in the revised manuscript.
>
> ---
>
> ## Comment 4: AR Baseline Architecture
>
> `For fairness, would the authors include an AR baseline with the same spatial backbone and time embedding that CFO uses, trained on the full grid?`
>
> We clarify our experimental design regarding architecture selection:
>
> * **For Main Results (Table 1):** To provide the strongest possible baseline, we evaluated AR across multiple architectures, including the **backbone used by CFO** (with zero-padding time). Table 1 reports the **best-performing AR variant** for each task. The results for the specific backbones are detailed in Appendix A.3 (Tables 4, 5, & 6).
> * **For Wall-Clock Comparisons (Table above):** We employed **identical spatial backbones** for both models. CFO uses sinusoidal time embeddings, while the AR baseline uses zero-padding to maintain an equivalent parameter count. This strictly isolates the performance gains stemming from the CFO training objective.
>
> ---
>
> ## Comment 5: Additional Harder Benchmark
>
> `It would be very valuable to add one harder PDE. Two natural choices are 2D Navier-Stokes (Kolmogorov flow) and Kuramoto-Sivashinsky, both available in community suites such as PDEBench.`
> **1D Kuramoto-Sivashinsky (KS) equation**
>
> We appreciate your suggestion. We evaluated CFO on the 1D Kuramoto-Sivashinsky (KS) equation from [5], a notoriously challenging chaotic PDE. All methods use the same 1D U-Net backbone; training details will be included in the appendix of the revised manuscript.
>
> | Method       | Rel. $L_2$ Error | Training Time|
> |--------------|---------------| ------
> | AR           | 0.317         | 1 h
> | Neural ODE   | 0.197         | 40 h 53 min
> | **CFO**      | **0.105**     | 1 h 5 min
>
> CFO achieves about **2× lower error** than the Neural ODE and **3× lower error** than AR. It is also much more **computationally efficient** than the Neural ODE baseline.
>
>
> References:
>
> [1] Zhang X N, Pu Y, Kawamura Y, et al. Trajectory flow matching with applications to clinical time series modelling[J]. Advances in Neural Information Processing Systems, 2024, 37: 107198-107224.
>
> [2] Kidger P, Morrill J, Foster J, et al. Neural controlled differential equations for irregular time series[J]. Advances in neural information processing systems, 2020, 33: 6696-6707.
>
> [3] Ricky T. Q. Chen, Yulia Rubanova, Jesse Bettencourt, and David Duvenaud.
>     “Neural Ordinary Differential Equations.” NeurIPS 2018.
>
> [4] Lippe, Phillip, et al. "Pde-refiner: Achieving accurate long rollouts with neural pde solvers." Advances in Neural Information Processing Systems 36 (2023): 67398-67433.
>
> [5] https://huggingface.co/datasets/hrrsmjd/kuramoto_sivashinsky

---

> ### Author Response · Authors · 2025-11-26
>
> Dear Reviewer,
>
> We are grateful for your detailed and constructive feedback. In our rebuttal, we have followed your suggestions by clarifying the method and experimental settings and by adding additional results.
>
> If there are any remaining questions, we would be happy to clarify them during the remaining discussion period, and we thank you for considering our responses in your final evaluation.

---

### Official Review · Reviewer_CPTa · 2025-10-28

**Soundness:** 2
**Presentation:** 3
**Contribution:** 2
**Rating:** 2
**Confidence:** 4

**Summary:**

The paper proposes a method to train spatial neural PDE models without relying on explicit integration-based optimization, as is common in neural ODE frameworks. Instead, temporal derivatives are supervised through stochastically interpolated trajectories derived from discrete training snapshots. The study further evaluates the method’s in-domain predictive performance within the same temporal horizon, under varying temporal sampling densities.

**Strengths:**

The main strengths of the paper are as follows:
- It proposes an approach that significantly reduces the computational and memory costs associated with applying directly neural ODE frameworks to PDE prediction tasks.
- It bridges flow learning and stochastic modeling by introducing a strategy to estimate temporal derivatives at unobserved time points through stochastic interpolation of training snapshots.

**Weaknesses:**

- Limited baselines: The paper only compares against a discrete-time autoregressive model. While neural ODE-based approaches can indeed be computationally demanding, this alone is not a sufficient reason to omit them from quantitative comparison. Including related continuous-time baselines such as the mentioned continuous-time methods (e.g., Chen et al., 2018 for ODE/PDEs; Yin et al., 2023 for PDEs) or other comparable approaches would strengthen the experimental validation.
- Unclear training details for autoregressive models: The paper does not specify how the autoregressive baselines are trained. Please provide more details about their training procedures (e.g., whether the model is rolled out for several time steps and supervised over the full trajectory).
- Different training strategies in practice similar to the proposed method: Common strategies such as teacher forcing (supervising one-step-ahead predictions) or temporal curriculum scheduling (progressively extending prediction horizons) can simplify training for both autoregressive and neural ODE models while reducing computational cost. These strategies can be viewed as alternative ways to achieve interpolation. A discussion or comparison of how these techniques relate to the proposed approach would provide valuable insight.
- Limited evaluation scope: The experiments focus solely on in-domain prediction. To better assess the model’s ability to learn underlying dynamics, the authors should evaluate extrapolation beyond the training horizon (e.g., training on data within [0, T/2) and testing on both [0, T/2) and [T/2, T)).

References
- Chen et al., 2018. Neural Ordinary Differential Equations. *NeurIPS 2018.*
- Yin et al., 2023. Continuous PDE Dynamics Forecasting with Implicit Neural Representations. *ICLR 2023.*

**Questions:**

See weakness.

---

> ### Author Response · Authors · 2025-11-21
> **Response to Reviewer CPTa (part 1)**
>
> ## Comment 1: Limited Baselines
>
> `The paper only compares against a discrete-time autoregressive model... including related continuous-time baselines such as Chen et al., 2018 for ODE/PDEs; Yin et al., 2023 for PDEs would strengthen the experimental validation.`
>
>
> We have added both **Neural ODE** [1] and **PDE-Refiner** [4] as baselines. As shown below, CFO significantly outperforms both in accuracy and training efficiency. Regarding latent-space methods (e.g., Yin et al., 2023), we note that such architectures are **orthogonal** to our contribution: they focus on **spatial** compression, whereas CFO introduces a novel **temporal** learning objective. CFO is compatible with latent representations and would likely achieve even higher efficiency when combined. Therefore, to isolate the impact of our temporal formulation, we prioritized comparisons against the direct continuous-time backbone (Neural ODE) on Lorenz and Burgers’ equations (low-dimensional, where latent representation is not strictly necessary), and the state-of-the-art spatiotemporal solver (PDE-Refiner) on high-dimensional 2D tasks.
>
> To ensure a fair comparison, we utilized identical backbone architectures and parameter counts under comparable experimental conditions. Training details will be provided in the appendix.
>
> 1. **Neural ODE Baseline**
> We implemented a Neural ODE baseline trained with:
> - Tsit5 [2] adaptive Runge-Kutta solver
> - `Diffrax`'s `RecursiveCheckpointAdjoint` [3] for memory-efficient backpropagation
> - Teacher forcing: one-step-ahead supervision minimizing $\|Solver(f_\theta,t_i,t_{i+1},u(t_i)) - u(t_{i+1})\|^2$, where $
> \mathrm{Solver}\bigl(f_\theta, t_i, t_{i+1}, u(t_i)\bigr)
> \approx
> u(t_i) + \int_{t_i}^{t_{i+1}} f_\theta\bigl(\tau, u(\tau)\bigr)\, d\tau
> $ denotes a numerical ODE solver applied to the neural vector field $f_\theta$ with the supervised initial condition $u(t_i)$.
>
>
> *Relative $L_2$ Error and Training Time per Batch:*
>
>
>
> | Equation | Method      | Rel. L₂ Error | Training Time (s/batch) |
> |----------|-------------|---------------|-------------------------|
> | Lorenz   | Neural ODE  | 0.101     | 0.133              |
> |          | **CFO**     | **0.0453** | **0.00350**          |
> | Burgers  | Neural ODE  | 0.0275     | 3.38                   |
> |          | **CFO**     | **0.00589** | **0.00920**          |
>
> CFO reduces relative error by 2.2 $\times$ on Lorenz and 4.7 $\times$ on Burgers' equation. Because CFO’s flow-matching objective avoids backpropagation through the ODE solver, it trains 38 $\times$ to 367 $\times$ faster per batch than the Neural ODE baseline.
>
> 2. **PDE-Refiner Baseline**
> Trained under the same settings and using the same backbone architecture with a matched parameter count, for 2D PDEs, we compare against **PDE-Refiner** [4], a state-of-the-art long-rollout PDE solver that incorporates diffusion-based denoising steps in the training and inference.
>
> *Relative $L_2$ Error and Training Time per Batch:*
>
> | Equation | Method       | Rel. L₂ Error | Training Time (s/batch) |
> |----------|--------------|---------------|-------------------------|
> | DR       | PDE-Refiner  | 0.125         | 1.38                   |
> |          | CFO          | **0.044**         | **0.40**        |
> | SWE      | PDE-Refiner  | 0.093         | 1.40                   |
> |          | CFO          | **0.005**         | **0.40**          |
>
> CFO reduces error by 2.8× on diffusion–reaction and 18.6× on shallow-water equations compared to PDE-Refiner. CFO is also 3–4× faster than PDE-Refiner because its denoising step (noise + step conditioning) makes each forward–backward pass more expensive than in our baseline.
>
>
> ---
>
> ## Comment 2: Unclear Training Details for Autoregressive Models
>
> `The paper does not specify how the autoregressive baselines are trained.`
>
> All autoregressive (AR) baselines use **teacher forcing**, the standard approach in time-dependent PDE literature.
>
> 1. **Training:** On discrete time grid $t_k = k\Delta t$, AR learns next-step map $u(t_{k+1}) \approx f_\theta(u(t_k))$ via one-step supervision:
>
> $$
> \mathcal{L} _ {\text{AR}}(\theta) = \frac{1}{N} \sum_{k=0}^{N-1} \|f_\theta(u(t_k)) - u(t_{k+1})\|^2_2
> $$
>
> 2. **Inference:** AR rollout from ground-truth initial condition $\hat{u}(t_0) = u_0$:
>
> $$
> \hat{u}(t_{k+1}) = f_\theta(\hat{u}(t_k)), \quad k = 0, \ldots, K-1
> $$
>
> This produces the entire trajectory $\{\hat{u}_1, \ldots, \hat{u}_N\}$ from the initial condition alone. We will add this clarification to the manuscript.
>
> ---

---

> ### Author Response · Authors · 2025-11-21
> **Response to Reviewer CPTa (part 2)**
>
> ## Comment 3: Different Training Strategies (Teacher Forcing, Curriculum)
>
> `Common strategies such as teacher forcing or temporal curriculum scheduling can simplify training... these strategies can be viewed as alternative ways to achieve interpolation.`
>
> We clarify that while these strategies simplify optimization, they do not achieve interpolation. The fundamental difference lies in the type of supervision used. We also show that these distinct supervision mechanisms lead to significantly different empirical results.
>
> 1. **Difference in supervision**.
> The key difference is the **supervision signal**:
> Both teacher forcing and temporal curriculum are supervised only at **discrete** training time points. However, CFO is supervised at **arbitrary continuous times** via a flow matching objective. The accuracy of this continuous supervision is controlled by finite-difference order and spline degree, enabling CFO to learn dynamics more precisely even from irregular and sparse data.
> All our baselines (Table 1 and the Neural ODE comparison in comment 1) already use **teacher forcing**, yet CFO still substantially outperforms them. CFO benefits from the continuous supervision.
>
> 2. **Temporal curriculum vs. CFO.**
> We additionally implemented  AR and neural ODE with a temporal curriculum, where the rollout horizon is gradually increased during training. This still requires simulating trajectories (multiple forward passes through time) during training and is therefore more expensive than pure one-step teacher forcing. It improves over vanilla AR and neural ODE on some equations but remains worse than CFO:
>
> *AR + Curriculum Results (Relative L₂ Error)*
> | Equation | AR + Curriculum | Quintic CFO (100%) |  Improvement |
> |----------|-----------------|--------------------|----------------------|
> | Lorenz   | 0.0709          | 0.0453             | 1.6×                 |
> | Burgers’ | 0.0137          | 0.00589            | 2.3×                 |
> | DR       | 0.3705          | 0.0437             | 8.5×                 |
> | SWE      | 0.0190          | 0.00456            | 4.2×                 |
>
> *Neural ODE + Curriculum Results (Relative L₂ Error)*
> | Equation | Neural ODE + Curriculum | Quintic CFO (100%) | Improvement |
> |----------|-----------------|--------------------|----------------------|
> | Lorenz   | 0.0801          | 0.0453             | 1.8×                 |
> | Burgers’ | 0.0174          | 0.00589            | 3.0×                 |
>
>
> The training details are provided in the appendix of the revised manuscript.
>
> 3. **Empirical Advantage**
>
> The difference leads to measurable benefits of CFO:
> - **Higher accuracy:** CFO consistently outperforms AR (with curriculum/teacher forcing) and Neural ODE (with curriculum/teacher forcing) (Table 1 and tables above).
> - **Faster convergence:** Training curves in the revised appendix show faster convergence for CFO.
> - **Data efficiency:** CFO at 25% irregular data outperforms AR at 100% uniform data (Table 1).
>
> Teacher forcing and temporal curriculum are therefore **orthogonal** to our contribution: CFO’s continuous-time, simulation-free formulation provides a fundamentally different and empirically stronger way to train neural PDE solvers.
>
> ---
>
> ## Comment 4: Limited Evaluation Scope (Extrapolation)
>
> `The experiments focus solely on in-domain prediction. The authors should evaluate extrapolation beyond the training horizon (e.g., training on [0, T/2), testing on [0, T)).`
>
> Excellent suggestion. We conducted extrapolation experiments training on the **first half** of trajectories [0, T/2) and evaluating on the full horizon [0, T].
>
> *Extrapolation Results (Relative L₂ Error)*
>
> | Equation | Train [0, T] | Train [0, T/2), Test [0, T] |
> |----------|--------------|----------------------------|
> | Lorenz   | 4.53×10⁻²    | 3.67×10⁻²                 |
> | Burgers  | 5.89×10⁻³    | 7.94×10⁻³                 |
> | DR       | 4.37×10⁻²    | 4.78×10⁻²                 |
> | SWE      | 4.56×10⁻³    | 2.28×10⁻²                 |
>
> We find that
>
> 1. **Lorenz, Burgers, DR:** Errors remain nearly identical, confirming CFO learns underlying dynamics rather than memorizing trajectories
>
> 2. **SWE:** Error increases to 2.28×10⁻² but remains **~4× better than AR baseline's in-domain performance** (9.04×10⁻²)
>
> These results demonstrate CFO's ability to generalize beyond the training temporal range, learning the governing dynamics rather than overfitting to specific time horizons. We will also include the error-accumulation curves in the revised manuscript.

---

> ### Author Response · Authors · 2025-11-21
> **Response to Reviewer CPTa (part 3)**
>
> References:
>
> [1] Ricky T. Q. Chen, Yulia Rubanova, Jesse Bettencourt, and David Duvenaud.
>     “Neural Ordinary Differential Equations.” NeurIPS 2018.
>
> [2] Ch. Tsitouras. “Runge–Kutta pairs of order 5(4) satisfying only the first column simplifying assumption.”
> Computers \& Mathematics with Applications, 62(2):770–775, 2011.
>
> [3] P. Kidger. Diffrax: numerical differential equation solvers in JAX. 2021.
>     URL: https://github.com/patrick-kidger/diffrax
>
> [4] Lippe, Phillip, et al. "Pde-refiner: Achieving accurate long rollouts with neural pde solvers." Advances in Neural Information Processing Systems 36 (2023): 67398-67433.

---

> ### Author Response · Authors · 2025-11-26
>
> Dear Reviewers,
>
> We appreciate your time and insightful comments on our submission. In our rebuttal, we have provided additional clarifications and results to address the concerns raised in the reviews.
>
> If there are any remaining questions, we would be happy to clarify them during the remaining discussion period, and we thank you for considering our responses in your final evaluation.

---

### Meta-Review · Area_Chair_FqyG · 2026-01-05

**Summary:**

The paper introduces a framework for learning continuous-time PDE dynamics. Using the method of lines, the PDE is first transformed into a system of ODEs. Sample trajectories of these ODEs are then approximated with spline representations by fitting both their values and their time derivatives, the latter being estimated via finite differences at discretized time steps. This construction yields a probabilistic path that approximates the velocity field governing the PDE dynamics. A neural operator is subsequently trained to regress this spline-based representation. The key claim is that, unlike NeuralODE-based approaches—which also enable continuous-time prediction but require differentiating through a numerical solver during training—the proposed method avoids backpropagation through the solver. Instead, training reduces to learning a neural operator regressor, resulting in improved computational efficiency. At inference time, starting from an initial condition, the learned operator is used to integrate the estimated velocity field and forecast the system trajectory. Experiments are conducted on several 1D and 2D benchmark PDEs. The main novelty lies in the use of a flow-matching loss to learn the velocity field, enabling continuous-time querying of the dynamics without reliance on solver differentiation.

The reviewers recognize the originality of repurposing flow matching for learning physical ODE dynamics. They raise several concerns, including limited baselines, a restricted set of benchmarks, and some imprecision in the technical description. In response, the authors provide a thorough rebuttal and add multiple new experiments, including evaluations on additional benchmark datasets and comparisons with new baselines.

**Reviewer Concerns:**

In my view, these additions adequately address the reviewers’ concerns. Given the extensive rebuttal and the inclusion of several new experiments with additional baselines and datasets that directly address the reviewers’ questions and concerns, I recommend acceptance.

**Reviewer Scores:**

RCPTa, rating 2, they adressed all the weaknesses and questions, adding new experiments that cover in my opinion the reviewer concerns, would probably raise their score

RCE6Q, rating 6, they answered all the questions/ concerns, would probably keep their score

Rweht, rating 2, extensive answer during the rebuttal, however I did not consider this review given its low-quality

R3Ms9, rating 4, adress all the concerns, add experiments on new benchmarks and baselines. The reviewer indicates that they will raise their score to 6.

---

### Decision · Program_Chairs · 2026-01-26

Accept (Poster)